# ARCHITECTURAL INSIGHTS FOR EFFICIENT PHYSICS-INFORMED NEURAL NETWORK OPTIMIZATION

## ABSTRACT

Physics-informed neural networks (PINNs) offer a promising avenue for tackling both forward and inverse problems in partial differential equations (PDEs) by incorporating deep learning with fundamental physics principles. Despite their remarkable empirical success, PINNs have garnered a reputation for their notorious training challenges across a spectrum of PDEs. In this work, we delve into the intricacies of PINN optimization from a neural architecture perspective. Leveraging the Neural Tangent Kernel (NTK), our study reveals that Gaussian activations surpass several alternate activations when it comes to effectively training PINNs. Building on insights from numerical linear algebra, we introduce a preconditioned neural architecture, showcasing how such tailored architectures enhance the optimization process. Our theoretical findings are substantiated through rigorous validation against established PDEs within the scientific literature.

## 1 INTRODUCTION

Physics-informed neural networks (PINNs) stand as a pioneering approach at the crossroads of deep learning and physics-based modeling. These innovative neural architectures seamlessly integrate the fundamental principles of physics into their learning process. By combining the predictive capabilities of neural networks with the governing equations that describe physical phenomena, PINNs facilitate the efficient and accurate modeling of complex partial differential equations (PDEs), even when data is limited or noisy. This innovative approach has demonstrated significant successes across a spectrum of domains, including fluid dynamics (Raissi et al., 2020; Sun et al., 2020; Jin et al., 2021), stochastic differential equations (Zhang et al., 2020), bioengineering (Sahli Costabal et al., 2020), and climate modeling (de Wolff et al., 2021; Zhu et al., 2022).

Practitioners in the field of Physics-Informed Neural Networks (PINNs) have consistently encountered a shared challenge: training instability leading to suboptimal predictions, particularly when dealing with physics problems exhibiting high-frequency solutions. This issue has also surfaced in the computer vision domain, where it is partly attributed to spectral bias—a tendency for neural networks to favor low-frequency solutions over higher-frequency ones Rahaman et al. (2019); Xu et al. (2022). To address this challenge, vision practitioners have adopted various techniques such as incorporating positional encodings (Tancik et al., 2020), sinusoidal activations (Sitzmann et al., 2020), Gaussian activations (Ramasinghe & Lucey, 2022; Chng et al., 2022; Saratchandran et al., 2023), and wavelet activations (Saragadam et al., 2023). While these approaches have demonstrated considerable success, a theoretical foundation explaining their superiority over classical architectures remains elusive. Furthermore, these innovative architectures are gaining traction in the PINN community (Wang et al., 2021; Faroughi et al., 2022; Zhao et al., 2023; Uddin et al., 2023), emphasizing the need for a deeper understanding of their mechanisms.

In this study, we provide a theoretical basis for the superior performance of Gaussian activations in fully connected architectures, using the Neural Tangent Kernel (NTK). Our findings indicate that in networks with a single wide layer of width $n_k$, linear in the number of training samples $N$ and adhering to a mild Lipschitz bound, the minimum eigenvalue of the empirical NTK exhibits a lower bound scaling of $\Omega(n_k^4)$. This compares favorably to previous research (Nguyen et al., 2021) showing a scaling of $\Omega(n_k)$ for ReLU-activated networks and a scaling of $\Omega(n_k^2)$ (Bombari et al., 2022) for non-linear Lipschitz activations with Lipschitz gradients, such as Tanh and sigmoid in networks with a pyramidal topology. To validate our insights, we conducted experiments with various PINN ar-

chitectures, consistently demonstrating the superior performance of Gaussian-activated PINNs over existing approaches commonly used in the field.

While addressing spectral bias in PINNs is crucial for achieving physically realistic predictions, it's worth noting, as highlighted in Wang et al. (2020; 2022), that the incorporation of physical constraints into the neural network's loss objective introduces significant challenges during optimization. This integration tends to adversely affect the smoothness of the loss landscape, often trapping gradient-based optimizers in local minima. To address this issue, we propose an architectural enhancement for feedforward PINNs, referred to as equilibrated PINNs. Equilibrated PINNs are specifically designed to improve the conditioning of the network's weight matrices during the optimization process, drawing inspiration from the concept of matrix preconditioning in numerical linear algebra (Chen, 2005). Our theoretical analysis sheds light on how this modification enhances the conditioning of the loss landscape, thereby facilitating more efficient training for gradient-based optimizers. We test our architecture against several PINN frameworks from the literature, across diverse PDEs, demonstrating that our architecture excels in various scenarios.

Our main contributions are:

1. We lay the theoretical groundwork for understanding the scaling pattern of the minimum eigenvalue of the empirical NTK of a Gaussian-activated neural network, thereby substantiating their suitability for PINN architectures. We then demonstrate the effectivness of our theory on various PDEs used within the PINN community.

2. We introduce an innovative PINN architecture that leverages matrix conditioning for the weights of a feedforward network during training. This approach leads to a more stable loss landscape and expedites the training process. Through experiments, we demonstrate the superior performance of this PINN architecture compared to several recently proposed counterparts across various benchmark PDEs commonly employed in the community.

## 2 RELATED WORK

**Optimization:** Initially, gradient-based optimization algorithms such as Adam and LBFGS were the go-to choices for training purposes, as seen in works by Raissi et al. Raissi et al. (2019) and Zhao et al. Zhao et al. (2023). To enhance convergence and efficiency, more advanced techniques like Bayesian and surrogate-based optimization have been harnessed (Yang et al., 2021). The application of regularization techniques, including learning rate annealing (Wang et al., 2020) and NTK regularization (Wang et al., 2022), has also exhibited notable effectiveness in improving training. In recent times, the field has witnessed progress with the introduction of meta-learning methods tailored specifically for the optimization of PINNs (Bihlo, 2023). The challenges faced during optimization of PINNs are often attributed to the ill-conditioned nature of the loss landscape, as underscored in studies by Fonseca et al. Fonseca et al. (2023), Basir et al. Basir & Senocak (2022), Wang et al. Wang et al. (2020). Multiple investigations have proposed that the main cause of this ill-conditioning lies in the training mismatch between the PDE residual and the boundary loss. In response, several researchers have experimented with re-weighting algorithms as a means to address and mitigate this mismatch (Xiang et al., 2022; Li & Feng, 2022; Hua et al., 2023; Batuwatta-Gamage et al., 2023; Deguchi & Asai, 2023).

**Activations and Architecture:** Numerous studies have explored diverse activation functions for training Physics-Informed Neural Networks (PINNs). Notably, Uddin et al. Uddin et al. (2023) and Zhao et al. Zhao et al. (2023) have investigated the use of wavelet activations, while Jagtap et al. Jagtap et al. (2020) have delved into locally adaptive activations. Furthermore, Faroughi et al. Faroughi et al. (2022) have employed periodic activations in their research. Recent advancements in PINN architecture have also demonstrated the potential for optimization improvements, such as the incorporation of positional embedding layers Wang et al. (2021) and the utilization of transformer methods Zhao et al. (2023).

## 3 GAUSSIAN ACTIVATIONS: INSIGHTS THROUGH THE NTK

### 3.1 BASICS ON PINNS

We consider PDEs defined on bounded domains $\Lambda \subseteq \mathbb{R}^n$. To this end, we seek a solution $u : \Lambda \to \mathbb{R}$ of the following system

$$\mathcal{N}[u](x) = f(x), x \in \Lambda \tag{1}$$
$$u(x) = f(x), x \in \partial\Lambda. \tag{2}$$

where $\mathcal{N}$ denotes a differential operator. In the setting of time-dependent problems, we will treat the time variable $t$ as an additional space coordinate and let $\Lambda$ denote the spatio-temporal domain. In so doing, we are able to treat the initial condition of a time-dependent problem as special type of Dirichlet boundary condition that can be included in equation 2.

The goal of physics informed neural network theory is to approximate the latent solution $u(x)$ of the above system by a neural network $u(x; \theta)$, where $\theta$ denotes the parameters of the network. The PDE residual is defined by $r(x; \theta) := u(x; \theta) - f(x)$. The key idea as presented in Raissi et al. (2019) is that the network parameters can be learned by minimizing the following composite loss function

$$\mathcal{L}(\theta) = \mathcal{L}_b(\theta) + \mathcal{L}_r(\theta) \tag{3}$$

where $\mathcal{L}_b$ denotes the boundary loss term and $\mathcal{L}_r$ denotes the PDE loss term, defined by

$$\mathcal{L}_b(\theta) = \frac{1}{2N_b} \sum_{i=1}^{N_b} |u(x_b^i; \theta) - g(x_b^i)|^2 \text{ and } \mathcal{L}_r(\theta) = \frac{1}{2N_r} \sum_{i=1}^{N_r} |r(x_r^i; \theta)|^2. \tag{4}$$

$N_b$ and $N_r$ represent the training points for the boundary and PDE residual. Minimizing both loss functions, $\mathcal{L}_b$ and $\mathcal{L}_r$, simultaneously using gradient-based optimization aims to learn parameters $\theta$ for an effective approximation, $u(x; \theta)$, of the latent solution, see Raissi et al. (2019).

In this work, we will also need to make use of some basic NTK theory for PINNs. We kindly ask the reader to consult appendix A.1 for a brief overview of the NTK and how it's applied for PINNs. The PINN loss function being a sum of two terms, $\mathcal{L}_b + \mathcal{L}_r$, gives rise to two main NTK terms $K_{uu}$, which corresponds to the boundary data, $K_{rr}$ which corresponds to the PDE data, and a mixed term $K_{ur}$, see appendix A.1 for an overview and Wang et al. (2022; 2021) for details.

Several works (Du et al., 2018; Allen-Zhu et al., 2019; Oymak & Soltanolkotabi, 2020; Nguyen & Mondelli, 2020; Zou & Gu, 2019) have established connections between the spectrum of the empirical NTK matrix and the training of neural networks. A key insight on this front is that when a neural network $u(x; \theta)$ is trained with Mean Squared Error (MSE) loss, $\mathcal{L}(\theta) = \frac{1}{2}||u(x; \theta) - y||_2^2$, where $y$ are the training labels, then it can be shown that

$$||\nabla\mathcal{L}(\theta)||_2^2 \geq 2\lambda_{\min}(K_{uu})\mathcal{L}(\theta) \tag{5}$$

The key take away is that **the larger $\lambda_{\min}(K_{uu})$ is at initialization the higher chance of converging to a global minimum**.

### 3.2 MOTIVATION

Consider the Poisson problem given by

$$-\Delta u = \pi^2 sin(\pi x) \text{ for } x \in [-1, 1] \tag{6}$$
$$u(-1) = u(1) = 0. \tag{7}$$

This problem poses both an existence and uniqueness challenge. Initially, equation 6 seeks a function satisfying $-\Delta u = \pi^2 \sin(\pi x)$, which admits multiple solutions, such as $u(x) = \sin(\pi x) + c$ with any constant $c$. To ensure uniqueness, equation 7 acts as a constraint, effectively singling out one unique solution from the family of solutions in equation 6, represented as $\sin(\pi x) + c; c \in \mathbb{R}$. Consequently, in the context of a PINN $u(x; \theta)$ with boundary loss $\mathcal{L}_b$ and residual loss $\mathcal{L}_r$, it becomes crucial to minimize the boundary loss $\mathcal{L}_b$ to near zero. Failure to do so could lead the PINN $u(x; \theta)$ to approximate one of the infinitely many residual solutions, like $sin(\pi x) + c$, where $c \neq 0$.

The above example highlights a common observation met by practitioners in the field. The loss functions $\mathcal{L}_b$ and $\mathcal{L}_r$ often decay to zero at very different rates for various PDEs Wang et al. (2022; 2020); Cuomo et al. (2022); Fuks & Tchelepi (2020) often causing the PINN to predict an incorrect solution. The main solution to this issue that has been investigated is loss re-weighting. Choosing parameters $\lambda_b$ and $\lambda_r$, a regularized loss of the form

$$\lambda_b \mathcal{L}_b(\theta) = \lambda_r \mathcal{L}_r(\theta) \tag{8}$$

is used to train the PINN. In general there have been many results suggesting methods to best choose $\lambda_b$ and $\lambda_r$ (Bischof & Kraus, 2021; Xiang et al., 2022; Wang et al., 2022; Perez et al., 2023). For example, Wang et al. (2022) employ an NTK approach to dynamically update $\lambda_b$ and $\lambda_r$ throughout training.

Our approach differs from standard methods by prioritizing architectural adjustments over traditional loss function regularization. We aim to maintain a large non-zero minimum eigenvalue for the boundary Neural Tangent Kernel ($K_{uu}$) during training, inspired by equation 5. Our empirical results suggest that choosing an architecture that maximizes this value at initialization significantly improves the likelihood of minimizing the boundary loss.

### 3.2.1 MAIN RESULT

In this section, we will demonstrate that Gaussian-activated feedforward PINNs (G-PINNs) exhibit a boundary NTK with a minimum eigenvalue that can be constrained to remain above a quartic term dependent on the width of a single wide layer.

The proof will require several assumptions that are common in the literature. Together with an added assumption on the Lipshitz constant of the network. Due to space constraints we have put the full general theorem together with all assumptions in appendix A.1. We will now give a synopsis of the theorem but kindly ask the reader to consult appendix A.1 for detailed notations, assumptions and proofs.

**Theorem 3.1.** *Let $u$ denote a depth $L$ neural network with $\phi(x) = e^{\frac{-x^2}{s^2}}$ as the activation, where $s^2 > 0$ is a fixed variance hyperparameter. Assume the first $L - 1$ widths $\{n_1, \ldots, n_{L-1}\}$ are all the same $\overline{N}$. Assume that $n_k \geq N \geq \overline{N}$ for $1 \leq k \leq L - 1$. Further assume the output dimension $n_L = 1$. Then*

$$\lambda_{\min}(K_{uu}) \geq \Omega\left(\frac{1}{s^3}\beta_k^4 n_k^4\right) \ w.h.p \ . \tag{9}$$

We pause here to remark that in Bombari et al. (2022) a more general result is proven. They prove that any Lipshitz non-linear activation with gradient having bounded Lipshitz constant has a quadratic scaling. There results imply that Gaussian-activated networks have an NTK whose minimum eigenvalue admits a qudratic scaling. The difference between our work and theirs is that we work in a highly overparameterized setting and assume a Lipshitz constant bound (see assumption A4 in appendix A.1). In doing so we are able to show how the lower bound of the mininmum eigenvalue of the NTK depends on the Lipshitz constant and the initialization. Our proofs also depend on the fact that we are employing a Gaussian activation and don't seem to go through for activations such as Tanh. However, Bombari et al. (2022) does apply to Tanh showing that the minimum eigenvalue of the NTK of a Tanh-activated network, in the asymptotic width setting, grows quadratically. While our assumptions seem more stringent to those in Bombari et al. (2022), we empirically verify our main assumption in appendix A.1. Furthermore, it should be noted that Nguyen et al. (2021) proved a linear scaling of the minimum eigenvalue for ReLU networks.

Fig. 1 empirically verifies the claim given by thm. 3.1 (see appendix A.1 for the more general statement). We compared two distinct 2-hidden layer networks, one employing a Gaussian activation of the form $e^{x^2/2(0.1)^2}$, and the other employing a Tanh activation. We fixed the widths of the input and output layers as $(n_0, n_2)$ and let the width of the middle layer, $n_1$, vary according to the relation $n_1 = 8N$. The Gaussian-activated network used a fixed variance parameter of $s^2 = 0.1^2$. Both networks were initialized using He's initialisation, where the weights $(W_l)_{i,j} \sim \mathcal{N}(0, \frac{2}{n_{l-1}})$. We then plotted the minimum eigenvalue of the empirical NTK $\lambda_{\min}(K_{uu})$ of both networks, and compared them to curves of the form $\mathcal{O}((8x)^4)$. As predicted by thm. 3.1, we observed that the minimum eigenvalue $\lambda_{\min}(K_{uu})$ grew faster than $\mathcal{O}((8x)^4)$. Furthermore we observed that the

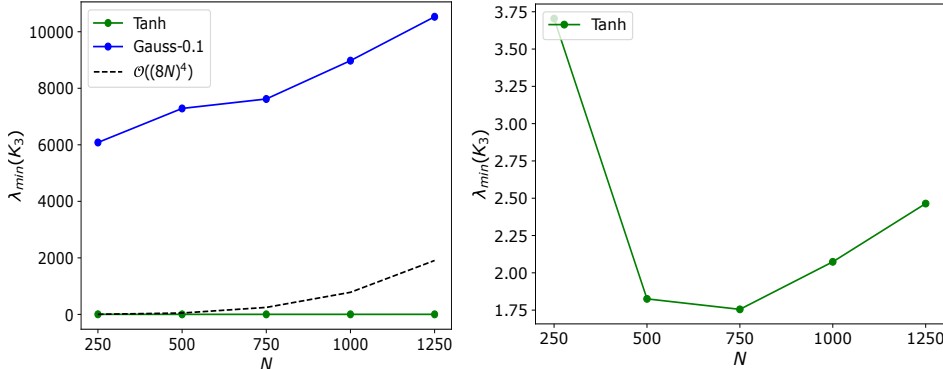

Figure 1: Right; The min. eigenvalue of the empirical NTK for a 2-hidden layer network. We took $N = 400$, $n_1 = 8N$, and $n_2 = 400$. As predicted by Thm. 3.1, $\lambda_{\min}(K_{uu})$ for a Gaussian-activated network grows much faster than a Tanh one. Left; zoom in of tanh network.

minimum eigenvalue of the NTK of the Gaussian network grew much faster than that of the Tanh network. Further experiments are carried out in appendix A.1.

## 4 PRECONDITIONED NEURAL ARCHITECTURES

Preconditioning is crucial in numerical solutions for linear systems $Ax = b$, aiming to lower the condition number of the matrix $A$. Reduced condition numbers lead to more accurate and efficient solutions. We remind the reader that the condition number of a matrix $A$ is defined as the ratio $\kappa(A) = \sigma_{\max}(A)/\sigma_{\min}(A)$, where $\sigma_{\min}$ and $\sigma_{\max}$ denotes the min and max singular values respectively. Typically, this process transforms the original system into a simpler one, denoted as $PAx = Pb$. The key challenge in preconditioning is selecting an appropriate multiplier $P$ to effectively reduce the initially high condition number to a significantly smaller value

We will concern ourselves with a particular form of preconditioning known as row equilibration, which involves taking the preconditioner $P$ to be $diag(||A_{i:}||_2)^{-1}$ i.e. $P$ is a diagonal matrix whose entries are the inverse of the row norms of $A$. Diagonal preconditioners are often used in numerical linear algebra as they are efficient to implement. Furthermore, the motivation to focus on row equilibration comes from the following theorem.

**Theorem 4.1** (Van der Sluis (1969)). *Let $A$ be a $n \times n$ matrix, $P$ an arbitrary diagonal matrix and $E$ the row equilibrated matrix built from $A$. Then $\kappa(EA) \leq \kappa(PA)$.*

Van Der Sluis' theorem highlights that among the set of diagonal preconditioners, row equilibration is the most effective for minimizing the condition number. For further insights into preconditioning, particularly quantitative results demonstrating the benefits of equilibration in reducing the condition number, we refer the reader to appendix A.3.

### 4.1 PRECONDITIONING THE LOSS LANDSCAPE

In this section, we examine an approach to conditioning the loss landscape through adjustments to the neural weights.

Consider a neural network $u(x; \theta)$ with $L$ layers, where the number of neurons in each layer are represented by $\{n_1, \ldots, n_L\}$. The feature maps, $u_k : \mathbb{R}^{n_0} \to \mathbb{R}^{n_k}$ for the network are defined for each input $x \in \mathbb{R}^{n_0}$ by

$$u_k(x) = \begin{cases} W_L^T f_{L-1} + b_L, & k = L \\ \phi(W_k^T u_{k-1} + b_k), & k = [L-1] \\ x, & k = 0 \end{cases} \tag{10}$$

where $n_0$ denotes the input dimension, $W_k \in \mathbb{R}^{n_{k-1} \times n_k}$, $b_k \in \mathbb{R}^{n_k}$, and $\phi$ is an activation, and the notation $[m] = \{1, \ldots, m\}$. The neural network $u(x; \theta)$ is then given by the composition of the layer maps $u_k$.

The MSE loss function associated to the neural network $u$ has Hessian given by

$$H(\mathcal{L}) = (Du)^T H(c) Du + (Dc) H(u) \tag{11}$$

where $D$ denotes the derivative with respect to parameters, $c$ is the qudratic convex cost function associated to the MSE loss function, see MacDonald et al. (2022) for details.

The matrix $(Du)^T H(c) Du$ is known as the Gauss-Newton matrix associated to $H(\mathcal{L})$ and is a common approximation to the Hessian $H(\mathcal{L})$ (Nocedal & Wright, 1999). We will use $(Du)^T H(c) Du$ as a surrogate to accessing $H(\mathcal{L})$. In fact, in Papyan (2019) it was shown that the large eigenvalues of $H(\mathcal{L})$ closely correlate to those of the Gauss-Newton matrix $(Du)^T H(c) Du$.

Using the chain rule we have

$$Du = \sum_{k=1}^{L} Ju_L \cdots Ju_{k+1} Du_k \tag{12}$$

where $Ju_i$ denotes the Jacobian of the $ith$ layer $u_i$ w.r.t inputs, and by the chain rule again we have

$$Ju_i = diag(\phi'(W_i^T u_{i-1} + b_i)) \cdot W_i \tag{13}$$

where $diag(\phi'(W_i^T u_{i-1} + b_i))$ denotes the diagonal matrix with diagonal entries $\phi'(W_i^T u_{i-1} + b_i)$.

For a PINN, $\mathcal{L} = \mathcal{L}_b + \mathcal{L}_r$, see equation 5. The Hessian is a linear operator, therefore $H(\mathcal{L} = H(\mathcal{L}_b) + H(\mathcal{L}_r)$. The Gauss-Newton matrix associated to $H(\mathcal{L}_r)$ now takes the form $(D\mathcal{N}(u))^T H(c) D\mathcal{N}(u)$, and using the chain rule as we did above, it is easy to see that $(D\mathcal{N}(u))^T H(c) D\mathcal{N}(u)$ will depend on the network weights of $u$.

By leveraging the property that the condition number of a product satisfies $\kappa(AB) \leq \kappa(A)\kappa(B)$, we find motivation to enhance the condition number of $(Du)^T H(c) Du$ and $(D\mathcal{N}(u))^T H(c) D\mathcal{N}(u)$ by improving the condition number of the weights $W_i$. While this isn't a formal proof, it provides the impetus to explore methods for reducing the condition number of $H(\mathcal{L})$. In the following section, we present quantitative techniques for achieving this.

## 4.2 Equilibrated PINNs

In this section we define an architecture that imposes row equilibration on the neural weights of a neural network.

Given a neural network $u$, as defined in sec. 4.1, we define an equilibrated network $u^E$ as follows: The layer maps of $u^E$ will be defined by:

$$u_k^E(x) = \begin{cases} W_L^T f_{L-1} + b_L, & k = L \\ \phi(P_k W_k^T u_{k-1}^E + b_k), & k = [2, L-1] \\ \phi(W_1 x + b_1), & k = 1 \\ x, & k = 0 \end{cases} \tag{14}$$

where each $P_k$ for $k \in [2, L-1]$ is the row equilibrated preconditioner (see sec. A.3 for definition) associated to $W_k^T$. $u^E$ is then given as the composition $u_L^E \circ \cdots \circ u_0^E$. Thus $u^E$ is obtained by equlibrating the inner weight matrices of the network. Note that the main reason we only equilibrate the inner weights is that we found empirically that this sufficed to yield good optimization.

Recall from equation 13, that the Jacobian of the ith layer (for $i > 0$) map $u_i$, w.r.t inputs, is given by $Ju_i = diag(\phi'(W_i^T u_{i-1} + b_i)) \cdot W_i^T$. We thus have

**Proposition 4.2.** $\kappa(Ju_i) \leq \kappa(diag(\phi'(W_i^T u_{i-1} + b_i)))\kappa(W_i^T)$.

By applying lems. A.14-A.16, we can generally conclude that $\kappa(P^i W_i^T)$ will be less than $\kappa(W_i^T)$, implying that the upper bound on $\kappa(Ju_i)$ as per prop. 4.2 will be lower for $\kappa(Ju_i^E)$. While not a rigorous proof, this observation strongly suggests that equilibrating the neural weights has the potential to contribute to improved conditioning of the loss landscape. We validate this insight through experiments in the next section and the appendix.

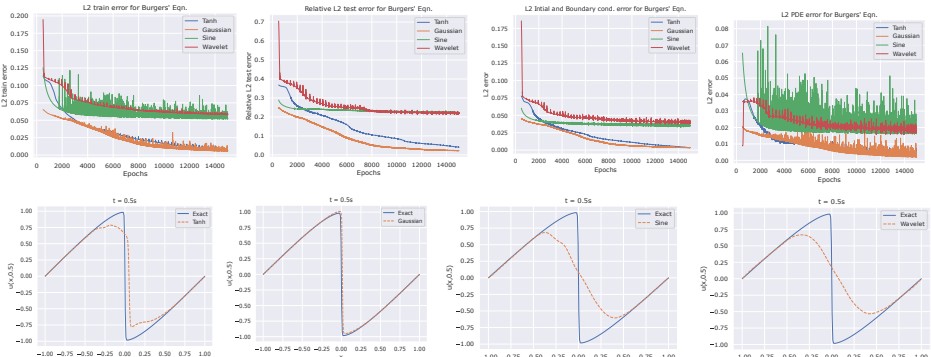

Figure 2: Top: Training/testing results for Burgers' equation. Bottom: Reconstruction of network solution plotted against exact solution at t= 0.5.

|  | $L^2$-train error | Rel. $L^2$-test error | Time (s) |
|---|---|---|---|
| $Tanh$ | $9.8e-3$ | $4.1e-2$ | $53.48$ |
| $Gaussian$ | $1.6e-3$ | $1.2e-2$ | $92.78$ |
| $Sine$ | $5.3e-2$ | $2.3e-1$ | $60.79$ |
| $Wavelet$ | $5.9e-2$ | $2.2e-1$ | $122.43$ |

Table 1: Train/test results for four different activated PINNs. Time denotes total training time to complete 15000 epochs.

## 5 EXPERIMENTS

### 5.1 BURGER'S EQUATION

We consider the Burger's equation, which is a convection-diffusion equation occurring in areas such as gas dynamics and fluid dynamics. The PDE takes the form

$$u_t + uu_x - \frac{0.01}{\pi}u_{xx} = 0 \text{ for } x \in [-1, 1] \text{ and } t \in [0, 1]. \tag{15}$$

$$u(x, 0) = -sin(\pi x) \tag{16}$$

$$u(-1, t) = u(1, t) = 0. \tag{17}$$

We aim to learn the solution, $u(x, t; \theta)$, using a PINN with training based on the loss defined in equation 3. In this experiment, we utilized 100 random points for initial and boundary data, along with 10000 points for PDE data. Four different PINNs were trained, each with distinct activations: Tanh Raissi et al. (2019), sine Faroughi et al. (2022), Gaussian, and wavelet Zhao et al. (2023). Further details regarding hyperparameter choices are given in the appendix A.4.

All PINNs used in the experiment featured 3 hidden layers with 128 neurons each and were trained using the Adam optimizer for 15000 epochs with a full batch of training points. Training/test results are shown in Table 1. Among them, the Gaussian PINN achieved the best performance. For a visual representation, refer to Figure 2, which illustrates the training curves and reconstructions at t = 0.5s. At this time point, the Gaussian PINN already provides a good approximation to the true solution, while other networks are struggling.

### 5.2 NAVIER-STOKES EQUATION

We consider the 2D incompressible Navier-Stokes equations as considered in Raissi et al. (2019).

$$u_t + uu_x + 0.01u_y = -p_x + 0.01(u_{xx} + u_{yy}) \tag{18}$$

$$v_t + uv_x + 0.01v_y = -p_y + 0.01(v_{xx} + v_{yy}) \tag{19}$$

|          | $L^2$ train error | Rel. $L^2$ test pressure error | Rel. $L^2$ test velocity error | Time (s) |
|----------|-------------------|--------------------------------|--------------------------------|----------|
| $Tanh$   | $2.9e-4$          | $2.8e-1$                       | $8.8e-5$                       | $887.48$ |
| $Gauss$  | $8.74e-5$         | $4.7e-2$                       | $3.4e-5$                       | $1466.10$ |
| $Sine$   | $2.0e-4$          | $6.8e-2$                       | $2.2e-4$                       | $958.18$ |
| $Wavelet$ | $5.0e-4$         | $1.2e-1$                       | $3.0e-4$                       | $1792.65$ |

Table 2: Training/testing results for the Navier-Stokes equations. The Gaussian-activated PINN achieves at least 2-4 times lower train/test errors than all other networks.

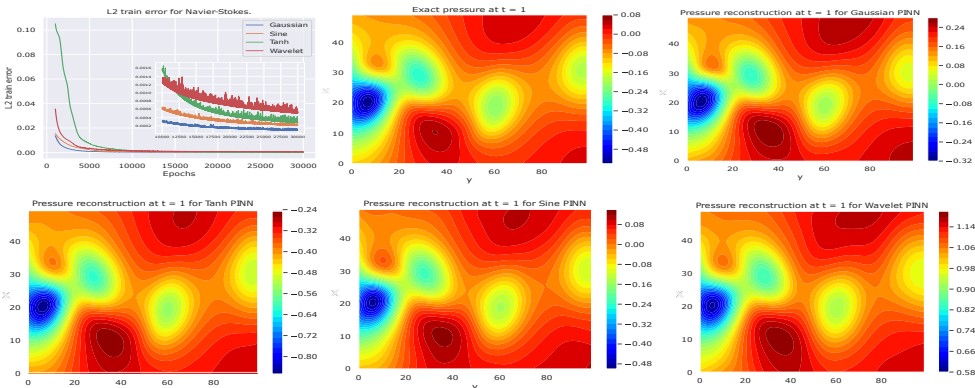

Figure 3: Top: $L^2$ train error (left). All other figures show reconstruction of the pressure field of each network at t = 1. The Gaussian-activated PINN has performed best in comparison to the others.

where $u(x, y, t)$ denotes the $x$-component of the velocity field of the fluid, and $v(x, y, t)$ denotes the $y$-component of the velocity field. The term $p(t, x, y)$ is the pressure. The domain of the problem is $[-15, 25] \times [-8, 8] \times [0, 20]$. We assume that $u = \psi_y$ and $v = -\psi_x$ for some latent function $\psi(t, x, y)$. With this assumption, the solution we seek will be divergence free, see Raissi et al. (2019) for details.

We trained four different PINNs to approximate a solution to the above system. The activations we tested were Tanh, Gaussian, sine, and wavelet. Each network had 3 hidden layers and 128 neurons in each layer. We used 5000 training data points consisting of training points for the velocity field and training points for the PDE residual. The networks were trained with an Adam optimizer for 30000 epochs.

Table 2 provides an overview of the training outcomes. It's evident that the Gaussian-activated PINN significantly outperforms other activation functions by a factor of at least 10. However, the training time of the Gaussian network is higher than sine and Tanh. In Fig. 3, we visualize the $L^2$ training error alongside the reconstruction of the pressure field $p(x, y, t)$ at $t = 1$. The superiority of the Gaussian network in approximating the field is evident when compared to the others. It's worth noting that the magnitudes of the reconstructed pressure fields for each PINN may differ from the exact pressure field, as the pressure field is only identifiable up to a constant Raissi et al. (2019).

### 5.3 HIGH FREQUENCY DIFFUSION

We consider a high frequency diffusion equation given by

$$u_t = u_{xx}(1 - (30\pi)^2)e^{-t}sin(30\pi x) \text{ for } x \in [-1, 1] \text{ and } t \in [0, 1] \quad (20)$$

$$u(x, 0) = sin(30\pi x) \text{ for all } x \in [-1, 1] \quad (21)$$

$$u(-1, t) = u(1, t) = 0 \text{ for all } t \in [0, 1]. \quad (22)$$

The system has a closed form analytic solution given by $u(x, t) = e^{-t}sin(30\pi x)$. We thus see that the solution diffuses in time but is high frequency in the space variable $x$. PINNs can struggle with finding an approximation to such a problem due to the high frequency spatial component.

|  | $L^2$ Train error | $L^2$ Rel. test error | $L^2$ Bndry./I.C. error | $L^2$ Pde error | Time (s) |
|---|---|---|---|---|---|
| G-PINN | $8.6e-3$ | $4.0e-1$ | $2.0e-4$ | $8.5e-3$ | 124.88 |
| EG-PINN | $2.2e-3$ | $2.9e-2$ | $7.55e-5$ | $3.4e-3$ | 439.22 |
| L-LAAF-PINN | $5.8e-3$ | $9.8e-2$ | $1.1e-2$ | $6.7e-3$ | 125.06 |
| PINNsformer | $1.1e7$ | $3.4e2$ | $8.5e1$ | $1.1e7$ | 17233.82 |
| RFF-PINN | $8.4e2$ | $1.1e0$ | $2.2e-1$ | $8.4e2$ | 125.35 |
| Tanh-PINN | $6.1e1$ | $1.5e4$ | $3.8e-1$ | $6.0e1$ | 76.86 |

Table 3: Training/Testing results for the diffusion equation. The first three entries all employ a Gaussian activation and are clearly superior to the other architectures yielding errors that are significantly lower than the others. Furthermore, amongst the 3 Gaussian-activated networks EG-PINN performs the best.

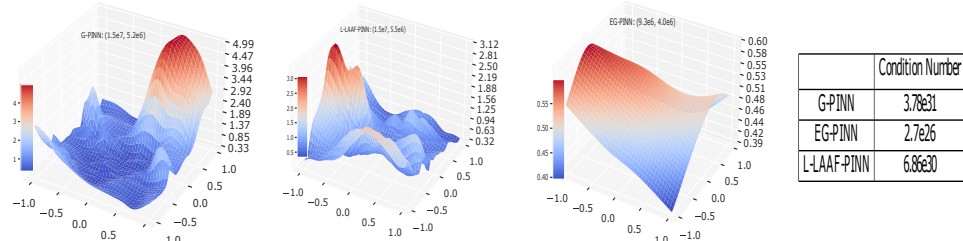

Figure 4: Loss landscape along the two most curved eigenvectors. The number at the top of each loss figure is the top two eigenvalues. Right: condition number of each network at that point. EG-PINN has a much lower condition number than the other two Gaussian-activated networks.

We tested various PINN architectures from the literature, including L-LAAF-PINN (Jagtap et al., 2020), PINNsformer with wavelet activation (Zhao et al., 2023), and RFF-PINN with Tanh activation (Wang et al., 2021). These were compared to two baseline architectures: Gaussian-PINN (G-PINN) and an equilibrated Gaussian activation architecture (EG-PINN), as described in Section 4.2. Notably, L-LAAF-PINN, which originally used swish activation, required switching to Gaussian activation for successful training on the given PDE. All architectures consisted of 3 hidden layers with 128 neurons, except for PINNsformer, which used the architecture from (Zhao et al., 2023) with 128 neurons in each of its 9 layers.

To substantiate the assertions in Section 4, we evaluated the condition numbers of PINNs, G-PINN, EG-PINN, and L-LAAF-PINN throughout their training processes. Notably, EG-PINN consistently maintained a significantly lower condition number. In Figure 4, we visually represent the loss landscape along the two most curved eigenvectors of the Hessian at an arbitrary point within the initial 10,000 epochs, emphasizing EG-PINN's smoother landscape and smaller condition number. Summarized in Table 3, the results highlight the exceptional performance of all three networks utilizing Gaussian activation, with EG-PINN surpassing the others. However, it's worth noting that EG-PINN requires considerably more training time than most of the other methods.

## 6 CONCLUSION AND LIMITATIONS

We demonstrated the utility of Gaussian activations in training neural networks by establishing a scaling law for the minimum eigenvalue of the empirical Neural Tangent Kernel (NTK) matrix. Applied to Physics-Informed Neural Networks (PINNs), this method enhances convergence of the boundary loss. Additionally, we introduced an architecture that conditions neural weights, smoothing the loss landscape for more effective optimization. However, limitations include our inability to establish a scaling law for the NTK related to the PDE residual, which could reveal differential convergence rates among PINN NTK terms. Moreover, the equilibrated architecture's computation of a row equilibration matrix after each backward pass extends training times, making it less practical for deeper and wider networks.

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

# A APPENDIX

## A.1 NTK

### A.1.1 A BRIEF REVIEW OF THE NTK

The Neural Tangent Kernel (NTK) is a pivotal concept in the realm of deep learning, offering a mathematical framework for understanding the training dynamics of neural networks. It quantifies how infinitesimal changes in a network's parameters relate to its output, playing a crucial role in analyzing and interpreting deep learning models. Originally introduced by Jacot et al. Jacot et al. (2018), the NTK has since garnered extensive attention and has been used to elucidate various aspects of neural network behavior.

For a neural network $u(x;\theta)$, the empirical NTK matrix over a batch of $N$ training data points $\{(x^i, y^i)\}$ is defined by

$$(K_{uu})_{ij} = \left\langle \frac{du(x^i;\theta)}{d\theta}, \frac{du(x^j;\theta)}{d\theta} \right\rangle \tag{23}$$

where $\left\langle \frac{du(x^i;\theta)}{d\theta}, \frac{du(x^j;\theta)}{d\theta} \right\rangle := \sum_\theta \frac{du(x^i;\theta)}{d\theta} \cdot \frac{du(x^j;\theta)}{d\theta}$.

Fix a physics informed neural network $u(x;\theta)$. We suppose that the training batches of the boundary points and PDE residual points are of size $N_b$ and $N_r$ respectively. It was shown in Wang et al. (2022) that the NTK of the PINN is composed of three major terms

$$(K_{uu})_{ij} = \left\langle \frac{du(x_b^i;\theta)}{d\theta}, \frac{du(x_b^j;\theta)}{d\theta} \right\rangle \tag{24}$$

$$(K_{ur})_{ij} = \left\langle \frac{du(x_b^i;\theta)}{d\theta}, \frac{d\sqcap(x_r^j;\theta)}{d\theta} \right\rangle \tag{25}$$

$$(K_{rr})_{ij} = \left\langle \frac{d\sqcap(x_b r^i;\theta)}{d\theta}, \frac{d\sqcap(x_r^j;\theta)}{d\theta} \right\rangle \tag{26}$$

The total NTK of equation 26, denoted $\mathbf{K}$, is then given by the block matrix

$$\mathbf{K} = \begin{bmatrix} K_{uu} & K_{ur} \\ K_{ur}^T & K_{rr}. \end{bmatrix} \tag{27}$$

We think of the term $K_{uu}$ as that part of the NTK that deals with the boundary condition, $K_{rr}$ as that part that deals with the PDE residual, and $K_{ur}$ the cross term dealing with how these interact.

In Wang et al. (2022) the NTK for a PINN was derived and the training of such networks was analyzed, yielding analogous results to those obtainined in Jacot et al. (2018). Furthermore, the NTK of PINNs has also been used to analyze Random Fourier Features (RFF) applied to PINN networks Wang et al. (2021).

### A.1.2 NOTATION AND ASSUMPTIONS

In this section we outline the notation and assumptions we will be using to prove the main theorem of the section, see thm. 3.1. We remark that the theorem we are going to prove is proven for general feed forward neural networks with a one dimensional output and thus is not restricted to PINNs. So as to avoid any confusion, in this section we will use the notation $f_L$ to denote our neural networks,

where $L$ is the depth, so that we can reserve the notation $u$ for when we want to work solely with a PINN. This should not create any issues and serves as a reminder to the reader that what we are proving is valid in the general context of feedforward networks.

**Notation:**  We will fix a depth $L$ neural network $f_L$, defined by [10] that admits a Gaussian activation of the form $e^{-x^2/2s^2}$. The data samples will be denoted by $X = [x_1, \ldots, x_N]^T \in \mathbb{R}^{N \times n_0}$, where $N$ is the number of samples and $n_0$ is the dimension of input features. The output of the network at layer $k$ will be denoted by $f_k$ and the feature matrix at layer $k$ by $F_k = [f_l(x_1), \ldots, f_l(x_N)]^T \in \mathbb{R}^{N \times n_k}$, where $n_k$ is the dimension of features at layer $k$. When $k = 0$, the feature matrix is simply the input data matrix $X$. We define $\Sigma_k = D([\phi'(g_{k,j}(x))]_{j=1}^{n_k})$, where $D$ denotes diagonal matrix and $g_{k,j}(x)$ denotes the pre-activation neuron i.e. the output of layer k before the activation function is applied. Note that $\Sigma_k$ is then an $n_k \times n_k$ diagonal matrix. We will use standard complexity notations, $\Omega(\cdot)$, $\mathcal{O}(\cdot)$, $o(\cdot)$, $\Theta(\cdot)$ throughout the paper, which are all to be understood in the asymptotic regime, where $N, n_0, n_1, \ldots, n_{L-1}$ are sufficiently large. Furthermore, we will use the notation "w.p." to denote *with probability*, and "w.h.p." to denote *with high probability* throughout the paper.

**Weight distributions:**  We will analyze properties of a network when weights are randomly initialized. Specifically, we assume that all weights are independently and identically distributed (i.i.d) according to a Gaussian distribution, $(W_k)_{i,j} \sim_{i.i.d} \mathcal{N}(0, \beta_k^2)$, where we assume $\beta_k \leq 1$ for all $1 \leq k \leq L - 1$.

**Data distribution:**  We will assume the data samples $\{x_i\}_{i=1}^N$ are i.i.d from a fixed distribution denoted by $\mathcal{P}$. The measure associated to $\mathcal{P}$ will be denoted by $d\mathcal{P}$. We will work under the following assumptions, which are standard in the literature.

A1. $\int_{\mathbb{R}^N} ||x||_2 d\mathcal{P} = \Theta(\sqrt{n_0})$.

A2. $\int_{\mathbb{R}^{n_0}} ||x||_2^2 d\mathcal{P} = \Theta(N)$.

We will also assume the data distribution satisfies Lipschitz concentration.

A3. For every Lipschitz continuous function $f : \mathbb{R}^{n_0} \to \mathbb{R}$, there exists an absolute constant $C > 0$ such that, for any $t > 0$, we have

$$\mathbf{P}\left(\left|f(x) - \int f(x')d\mathcal{P}\right|\right) \leq 2e^{-ct^2/||f||_{Lip}^2}.$$

Note that there are several distributions satisfying assumption A3 such as standard Gaussian distributions, and uniform distributions on spheres and cubes, see Vershynin (2018).

**Lipshitz Assumption:**  The final assumption we will be making is a Lipschitz constant assumption on the network.

A4. The Lipschitz constant of layer $k$ must satisfy the following bound

$$||f_k||_{Lip}^2 = \mathcal{O}\left(\frac{\beta_k n_k}{s^k \min_{l \in [0,k]} n_l}\left(\prod_{l=1}^{k-1} \sqrt{\beta_l}\sqrt{n_l}\right)\left(\prod_{l=1}^{k-1} Log(n_l)\right)\right).$$

### A.1.3 Proof of Main NTK theorem

We will prove a more general theorem than that of thm. 3.1.

**Theorem A.1.** *Let $f_L$ denote a depth $L$ neural network with $\phi(x) = e^{\frac{-x^2}{s^2}}$ as the activation, where $s > 0$ is a fixed frequency parameter, satisfying the network assumptions in Section Let $\{x_i\}_{i=1}^N$ denote a set of i.i.d training data points sampled from the distribution $\mathcal{P}$, which satisfies the data assumptions in Section Let $a_k = 1$ if the following conditions holds*

$$n_k \geq N$$

$$\prod_{l=1}^{k-2} Log(n_l) = o\left(\min_{l \in [0,k-1]} n_l\right)$$

*and* $0$ *otherwise. Then*

$$\lambda_{\min}(K_L) \geq \sum_{k=1}^{L-1} a_k \Omega \left( \frac{n_0^{\frac{3k+4}{2}} \beta_{k+1}^3 \beta_k^4 n_k^4}{s^3} \left( \prod_{l=1}^{k-1} \beta_l^{7/2} n_l^{7/2} \right) \left( \prod_{l=k+1}^{L} \beta_l n_l \right) \right)$$

$$+ \lambda_{min}(XX^T) \cdot \Omega \left( \frac{\beta_1^3 n_0^{3/2}}{s^3} \prod_{l=1}^{L} \beta_l n_l \right)$$

*w.p. at least*

$$1 - \sum_{k=1}^{L-1} (N^2 - N) \exp \left( -\Omega \left( \frac{s^k \min_{l \in [0,k-1]} n_l}{N^2 \prod_{l=1}^{k-2} Log(n_l)} \right) \right) - N \sum_{l=0}^{k} 2 \exp(-\Omega(n_l))$$

*over* $(W_l)_{l=1}^L$ *and the data.*

We note that as the widths approach infinity. The probability estimate at the end of the theorem goes to zero.

The proof of the theorem will follow the techniques of Nguyen et al. (2021); Bombari et al. (2022); Nguyen & Mondelli (2020). However, we want to iterate that there are several places where new additions and techniques must be carried out. In particular, expectation integrals will depend heavily on Gaussian analysis. When we use results from those papers we will explicitly say so by referencing them.

The theorem will be proven via a string of lemmas. To begin with we go through the basic idea of how we prove the theorem.

Recall that the empirical NTK as

$$K_L = JJ^T = \sum_{l=1}^{k} \left( \frac{\partial F_L}{\partial vec(W_l)} \right) \left( \frac{\partial F_L}{\partial vec(W_l)} \right)^T.$$

Using the chain rule, we can express the NTK in terms of the feature matrices as:

$$JJ^T = \sum_{k=0}^{L-1} F_l F_l^T \circ G_{k+1} G_{k+1}^T, \tag{28}$$

where $G_k \in \mathbb{R}^{N \times n_k}$ with $i^{th}$ row given by

$$(G_k)_{i:} = \begin{cases} \frac{1_N}{\sqrt{N}}, & k = L \\ \Sigma_{L-1}(x_i) W_L, & k = L-1 \\ \Sigma_k(x_i) \left( \prod_{j=k+1}^{L-1} W_j \Sigma_j(x_i) \right) W_L, & k \in [L-2] \end{cases}$$

By Weyl's inequality, we obtain

$$\lambda_{\min}(JJ^T) \geq \sum_{k=0}^{L-1} \lambda_{\min}(F_k F_k^T \circ G_{k+1} G_{k+1}^T). \tag{29}$$

Each term in the sum on the left hand side of equation 29 can be further bounded by Schur's Theorem Schur (1911); Horn et al. (1994) to give

$$\lambda_{\min}(F_k F_k^T \circ G_{k+1} G_{k+1}^T)$$
$$\geq \lambda_{\min}(F_k F_k^T) \min_{i \in [N]} ||(G_{k+1})_{i:}||_2^2. \tag{30}$$

equation 29 and equation 30 imply

$$\lambda_{\min}(JJ^T) \geq \sum_{k=0}^{L-1} \lambda_{\min}(F_k F_k^T) \min_{i \in [N]} ||(G_{k+1})_{i:}||_2^2. \tag{31}$$

The strategy of the proof is to obtain bounds on the terms $\lambda_{\min}(F_k F_k^T)$ and $||(G_{k+1})_{i:}||_2^2$ separately, and then combine them together to obtain a bound on $J J^T$. The following lemma shows how to estimate the quantity $||(G_{k+1})_{i:}||_2^2$.

**Lemma A.2.** *Fix $k \in [L-2]$ and let $x \sim \mathcal{P}$. Then*

$$\left\| \Sigma_k(x) \left( \prod_{l=k+1}^{L-1} W_l \Sigma_l(x) \right) W_L \right\|_2^2 = \Theta \left( \frac{\beta_k^3 n_0^{3k/2}}{s^3} \left( \prod_{l=1}^{k-1} \beta_l^3 n_l^3 \right) \left( \beta_l n_l \right) \right)$$

*w.p. at least $1 - \sum_{l=0}^{L-1} 2 \exp(-\Omega(n_l))$.*

The following theorem derives bounds on the minimum singular value of the feature matrices associated to the network. We consider a set of i.i.d data samples $\{x_i\}_{i=1}^N$, drawn from distribution $\mathcal{P}$, which is assumed to satisfy assumptions A1-A3. The feature matrix at layer $k$ is given by $F_k = [f_k(x_1), \ldots . f_k(x_N)]^T \in \mathbb{R}^{N \times n_k}$. For the following theorem, we assume assumption A4.

**Theorem A.3.** *Let $f_L$ denote a neural network of depth $L$ with activation function $\phi(x) = e^{-x^2/s^2}$, where $s^2 > 0$ is a fixed variance hyperparameter. We assume the following conditions hold:*

$$n_k \geq N$$

$$\prod_{l=1}^{k-1} Log(n_l) = o \left( \min_{l \in [0,k]} n_l \right).$$

*The minimum singular value of the feature matrix $F_k$, denoted $\sigma_{\min}(F_k)$, satisfies the following bound*

$$\sigma_{\min}(F_k)^2 = \Theta \left( \sqrt{n_0} \beta_k n_k \prod_{l=1}^{k-1} \sqrt{\beta_l} \sqrt{n_l} \right)$$

*w.p. at least*

$$1 - N(N-1) \exp \left( -\Omega \left( \frac{s^k \min_{l \in [0,k-1]} n_l}{N^2 \prod_{l=1}^{k-2} Log(n_l)} \right) \right) - N \sum_{l=1}^{k} 2 \exp(-\Omega(n_l))$$

The proof of theorem A.1 now follows from lemma A.2 and theorem A.3.

***Proof of theorem A.1***. Apply Theorem A.3 and lemma A.2 to obtain lower bounds on the terms in the sum on the right hand side of equation 31. $\qquad\square$

We are left with proving lemma A.2 and theorem A.3. In order to do this we will need a string of preliminary lemmas.

### A.1.4 PRELIMINARY LEMMAS

**Lemma A.4.** *Let $\phi$ denote the activation function $\phi(x) = e^{-x^2/s^2}$ where $s$ is a fixed standard deviation parameter. Fix $0 \leq k \leq L-1$ and assume $x \sim \mathcal{P}$. Then*

$$||f_k(x)||_2^2 = \Theta \left( s \cdot \sqrt{n_0} \prod_{l=1}^{k-1} \sqrt{\beta_l} \sqrt{n_l} \beta_k n_k \right)$$

*w.p. $\geq 1 - \sum_{l=1}^{k} 2exp(-\Omega(sn_l)) - 2exp(-\Omega(\sqrt{n_0}))$ over $(W_l)_{l=1}^k$ and $x$.*

*Furthermore*

$$\mathbb{E}_{x \sim \mathcal{P}} ||f_k(x)||_2^2 = \Theta \left( s \cdot \sqrt{n_0} \prod_{l=1}^{k-1} \sqrt{\beta_l} \sqrt{n_l} \beta_k n_k \right)$$

*w.p. $\geq 1 - \sum_{l=1}^{k} 2exp(-\Omega(sn_l))$ over $(W_l)_{l=1}^k$.*

*Proof.* The proof will be by induction. From the data assumptions, it is clear that the lemma is true for $k = 0$. Assume the lemma holds for $k - 1$, we prove it for $k$. The proof proceeds by conditioning on the event $(W_l)_{l=1}^{k-1}$ and obtaining bounds over $W_k$. Then by the induction hypothesis and intersecting over the two events the result will follow.

We have that $W_k \in R^{n_{k-1} \times n_k}$, so we can write $W_k = [w_1, \ldots, w_{n_k}]$, where each $w_i \in \mathbb{R}^{n_{k-1}}$ and $w_i \sim \mathcal{N}(0, \beta_k^2 I_{n_{k-1}})$. We then estimate

$$||f_k(x)||_2^2 = \sum_{i=1}^{k} ||f_{k,i}^2||^2.$$

Taking the expectation, we have

$$\mathbb{E}_{W_k} ||f_k||_2^2 = \sum_{i=1}^{k} \mathbb{E}_{w_i}[f_{k,i}(x)^2], \text{ by independence}$$

$$= n_k \mathbb{E}_{w_i}[f_{k,i}(x)^2].$$

By definition $f_{k,j}(x) = \phi(\langle w_j, f_{k-1}(x)\rangle)$. Note that the random variable $\langle w_j, f_{k-1}(x)\rangle$ is a univariate random variable distributed according to $\mathcal{N}(0, \beta_k^2 ||f_{k-1}(x)||_2^2)$. Computing this expectation comes down to computing the following integral $\int_{\mathbb{R}} e^{-2w^2/s^2} e^{\frac{-w^2}{\beta_k^2 ||f_{k-1}(x)||_2^2}} dw$. This integral can be computed from Gaussian integral computations as follows:

$$\int_{\mathbb{R}} e^{-2w^2/s^2} e^{\frac{-w^2}{\beta_k^2 ||f_{k-1}(x)||_2^2}} dw = \int_{\mathbb{R}} e^{-\left(\frac{2}{s^2} + \frac{1}{\beta_k^2 ||f_{k-1}||_2^2}\right) w} dw$$

$$= \beta_k ||f_{k-1}(x)||_2 s \sqrt{\frac{1}{2\beta_k^2 ||f_{k-1}||_2^2 + s^2}}.$$

Thus we get

$$\mathbb{E}_{w_i}[f_{k,i}(x)^2] = s \left(\frac{1}{2\beta_k^2 ||f_{k-1}||_2^2 + s^2}\right)^{1/2} \beta_k ||f_{k-1}||_2.$$

This in turn implies that

$$C \leq \mathbb{E}_{w_i}[f_{k,i}(x)^2] \leq \beta_k ||f_{k-1}||_2$$

for some constant $C > 0$. In particular, by induction we get

$$\mathbb{E}_{W_k}[||f_k(x)||_2^2] = \Theta\left(\sqrt{n_0} s \prod_{l=1}^{k-1} \sqrt{\beta_l} \sqrt{n_l} \beta_k n_k\right).$$

Using the above expectation we would like to apply Bernstein's inequality Vershynin (2018) to obtain a bound on $||f_k||_2^2$. In order to do this we need to compute the sub-Gaussian norm $||f_{k,j}(x)^2||_{\psi_1} = ||f_{k,j}(x)||_{\psi_2}^2$. Since $|e^{-x^2/s^2}| \leq 1$, we have

$$\mathbb{E}_{w_i}\left(exp\left(\frac{f_{k,j}(x)^2}{t^2}\right)\right) \leq \mathbb{E}_{w_j}\left(exp\left(\frac{1}{t^2}\right)\right) = exp\left(\frac{1}{t^2}\right) \beta_k^{n_k}. \tag{32}$$

By taking $t = \dfrac{1}{Log\left(\frac{1}{\beta_k^{n_{k-1}}}\right)}$ we get that

$$\mathbb{E}_{w_i}\left(exp\left(\frac{f_{k,j}(x)^2}{t^2}\right)\right) \leq \mathbb{E}_{w_j}\left(exp\left(\frac{1}{t^2}\right)\right) \leq 1,$$

using the fact that $\beta_k \leq 1$. In particular, we get that $||f_{k,j}(x)||_{\psi_2}^2 \leq \mathcal{O}(1)$.

Applying Bernstein's inequality Vershynin (2018) to $\sum_{i=1}^{n_k} \left(f_{k,i}(x)^2 - \mathbb{E}_{w_i}[f_{k,i}(x)^2]\right)$ we obtain

$$|\sum_{i=1}^{n_k} \left(f_{k,i}(x)^2 - \mathbb{E}_{w_i}[f_{k,i}(x)^2]\right)| \leq \frac{1}{2} \mathbb{E}_{W_k} ||f_k(x)||_2^2 \tag{33}$$

w.p $\geq 1 - 2exp(-c\frac{\mathbb{E}_{W_k}||f_k(x)||_2^2}{2})$. Thus we find that

$$\frac{1}{2}\mathbb{E}_{W_k}||f_k(x)||_2^2 \leq ||f_k(x)||_2^2 \leq \frac{3}{2}\mathbb{E}_{W_k}||f_k(x)||_2^2 \tag{34}$$

w.p. $\geq 1 - 2exp(-2s\Omega(n_k))$. Taking the intersection of the induction over $(W_l)_{l=1}^{k-1}$ and the even over $W_k$ proves the first part of the lemma.

The proof for $\mathbb{E}_x||f_k(x)||_2^2$ follows a similar argument using Jensen's inequality $||\mathbb{E}_x[f_{k,i}(x)^2]||_{\psi_1} \leq \mathbb{E}_x||f_{k,i}(x)^2||_{\psi_1} = \mathcal{O}(1)$.

$\square$

**Lemma A.5.** *Let $\phi(x) = e^{-x^2/s^2}$. Then*

$$||\mathbb{E}_x[f_k(x)]||_2^2 = \Theta(n_0 \prod_{l=1}^{l} \beta_l^2 n_l)$$

*w.p. $\geq 1 - \sum_{l=1}^{k} 2exp(-\Omega(n_l))$ over $(W_l)_{l=1}^{k}$.*

*Proof.* By Jensen's inequality $||\mathbb{E}_x[f_k(x)]||_2^2 \leq \mathbb{E}_x||f_k(x)||_2^2$. Thus the upper bound follows from lemma A.4.

The proof of the lower bound follows by induction. The $k = 0$ case following from the data assumption. Assume

$$||\mathbb{E}_x[f_k(x)]||_2^2 = \Omega(sn_0 \prod_{l=1}^{k-1} \beta_k)$$

w.p. $\geq 1 - \sum_{l=1}^{k-1} exp(-\Omega(n_l))$ over $(W_l)_{l=1}^{k-1}$. We condition on the intersection of this event and the event of lemma A.4 for $(W_l)_{l=1}^{k-1}$.

Write $W_k = [w_1, \ldots, w_{n_k}]$ with $w_j \sim \mathcal{N}(0, \beta_k^2 I_{n_k-1})$ for $1 \leq j \leq n_k$. Then

$$\begin{aligned}||\left(\mathbb{E}_x[f_{k,i}(x)]\right)^2||_{\psi_1} &= ||\mathbb{E}_x[f_{k,i}(x)]||_{\psi_2}^2 \\ &\leq \mathbb{E}_x||f_{k,i}(x)||_{\psi_2}^2 \\ &\leq C\sqrt{d}\end{aligned}$$

for some $C > 0$.

Moreover,

$$\begin{aligned}\mathbb{E}_{W_k}||\mathbb{E}_x[f_{k,i}(x)]||_2^2 &= \sum_{i=1}^{n_k} \mathbb{E}_{w_i}(\mathbb{E}_x[f_{k,i}(x)])^2 \\ &\geq \sum_{i=1}^{n_k}(\mathbb{E}_x\mathbb{E}_{w_i}[f_{k,i}(x)])^2 \\ &\geq \frac{\beta_k^2 n_k}{4}(\mathbb{E}_x||f_{k-1}(x)||_2)^2 \\ &= \Omega(sn_0 \prod_{l=1}^{k} \beta_l^2 n_l)\end{aligned}$$

where the second inequality is computed using the same technique as in lemma A.4.

Applying Bernstein's inequality Vershynin (2018) we get

$$||\mathbb{E}_x[f_k(x)]||_2^2 \geq \frac{1}{2}\mathbb{E}_{W_k}||\mathbb{E}_x[f_k(x)]||_2^2 = \Omega(sn_0 \prod_{l=1}^{n_k} \beta_l^2 n_l)$$

w.p. $\geq 1 - 2exp(-\Omega(n_k))$ over $(W_k)$. Taking the intersection of all the events then finishes the proof. $\square$

**Lemma A.6.** *Let $\phi(x) = e^{-x^2/s^2}$. Then for any $k \in [L-1]$ and any $i \in [N]$, we have*

$$||f_k(x_i) - \mathbb{E}_x[f_k(x)]||_2^2 = \Theta\left(\sqrt{n_0}\beta_k n_k \prod_{l=1}^{k-1}\sqrt{\beta_l}\sqrt{n_l}\right)$$

*w.p.* $\geq 1 - N\exp\left(-\Omega\left(\frac{\min_{l\in[0,k]} n_l}{\prod_{l=1}^{k-1} \log(n_l)}\right)\right) - \sum_{l=1}^{k} \exp(-\Omega(n_l))$.

*Proof.* Let $X : \mathbb{R}^{n_0} \to \mathbb{R}$ denote the random variable defined by $X(x_i) = ||f_k(x_i) - \mathbb{E}_x[f_k(x)]||_2$. By assumption A4, we have

$$||X||_{Lip}^2 = \mathcal{O}\left(\frac{\beta_k n_k \prod_{l=1}^{k-1}\sqrt{\beta_l}n_l \prod_{l=1}^{k-1}\text{Log}(n_l)}{s^k \min_{l\in[0,k]} n_l}\right)$$

w.p. $\geq 1 - \sum_{l=1}^{k} \exp(-\Omega(n_l))$.

We use the notation $\mathbb{E}[X] = \mathbb{E}_{x_i}[X(x_i)] = \int_{\mathbb{R}^{n_0}} X(x_i) d\mathcal{P}(x_i)$. We then have

$$
\begin{aligned}
\mathbb{E}[X]^2 &= \mathbb{E}[X^2] - \mathbb{E}[|X - \mathbb{E}X|^2] \\
&\geq \mathbb{E}[X^2] - \int_0^\infty \mathbb{P}(|X - \mathbb{E}X| > \sqrt{t})dt \\
&\geq \mathbb{E}[X^2] - \int_0^\infty 2\exp\left(\frac{-ct}{||X||_{Lip}^2}\right)dt \\
&= \mathbb{E}[X^2] - \frac{2}{c}||X||_{Lip}^2.
\end{aligned}
$$

By lemma A.7, we have w.p. $\geq 1 - \sum_{l=1}^{k} \exp(-\Omega(n_l))$ over $(W_l)_{l=1}^{k}$ that

$$\mathbb{E}[X^2] = \Theta\left(\sqrt{n_0}\beta_k n_k \prod_{l=1}^{k-1}\sqrt{\beta_l}\sqrt{n_l}\right)$$

which implies

$$\mathbb{E}[X] = \Omega\left(\sqrt{\sqrt{n_0}\beta_k n_k \prod_{l=1}^{k-1}\sqrt{\beta_l}\sqrt{n_l}}\right).$$

Moreover, by Jensen's inequality $\mathbb{E}[X] \leq \sqrt{\mathbb{E}[X^2]} = \mathcal{O}\left(\sqrt{\sqrt{n_0}\beta_k n_k \prod_{l=1}^{k-1}\sqrt{\beta_l}\sqrt{n_l}}\right)$.

Putting the above two asymptotic bounds together we obtain

$$\mathbb{E}[X] = \Theta\left(\sqrt{\sqrt{n_0}\beta_k n_k \prod_{l=1}^{k-1}\sqrt{\beta_l}\sqrt{n_l}}\right)$$

w.p. $\geq 1 - \sum_{l=1}^{k} \exp(-\Omega(n_l))$ over $(W_l)_{l=1}^{k}$.

We condition on the above event and obtain bounds over each sample. Using Lipschitz concentration, see assumption A3, we have that $\frac{1}{2}\mathbb{E}[X] \leq X \leq \frac{3}{2}\mathbb{E}[X]$. Therefore,

$$X = \Theta\left(\sqrt{\sqrt{n_0}\beta_k n_k \prod_{l=1}^{k-1}\sqrt{\beta_l}\sqrt{n_l}}\right)$$

w.p. $\geq 1 - \exp\left(-\Omega\left(\frac{\min_{l\in[0,k]} n_l}{\prod_{l=1}^{k-1} \log(n_l)}\right)\right)$. Taking the union bounds over the $N$ samples and intersecting them with the above event over $(W_l)_{l=1}^{k}$ gives the lemma. $\square$

**Lemma A.7.** *Let $\phi(x) = e^{-x^2/s^2}$. Then*

$$\mathbb{E}_x||f_k(x) - \mathbb{E}_x[f_k(x)]||_2^2 = \Theta\left(\sqrt{n_0}\beta_k n_k \prod_{l=1}^{k-1}\beta_l n_l\right)$$

*w.p. $\geq 1 - \sum_{l=1}^{k} exp(-\Omega(n_l))$ over $(W_l)_{l=1}^{k}$.*

*Proof.* The proof is by induction. Note that the $k = 0$ case is given by the concentration inequality assumption of the data. Assume the lemma is true for $k - 1$. We condition on this event over $(W_l)_{l=1}^{k-1}$ and obtain bounds over $W_k$. Then taking the intersection of the two events we will get a proof of the lemma.

We recall that we write $W_k = [w_1, \ldots, w_{n_k}]$ where $w_i \sim \mathcal{N}(0, \beta_k^2 I_{n_{k-1}})$. By expanding the squared norm we have

$$\mathbb{E}_x||f_k(x) - \mathbb{E}_x[f_k(x)]||_2^2 = \sum_{j=1}^{n_k}\mathbb{E}_x(f_{k,j}(x) - \mathbb{E}_x[f_{k,j}(x)])^2.$$

We now take the expectation over $W_k$ to obtain

$$\mathbb{E}_{W_k}\mathbb{E}_x||f_k(x) - \mathbb{E}_x[f_k(x)]||_2^2 = \mathbb{E}_{W_k}\mathbb{E}_x||f_k(x)||_2^2 - \mathbb{E}_{W_k}||\mathbb{E}_x f_k(x)||_2^2.$$

From the proof of lemma A.4, we know that

$$\mathbb{E}_{W_k}||f_k(x)||_2^2 \geq C\frac{\beta_k n_k}{2}||f_{k-1}(x)||_2$$

for some constant $C > 0$. Therefore, we can estimate

$$\mathbb{E}_{W_k}\mathbb{E}_x||f_k(x)||_2^2 - \mathbb{E}_{W_k}||\mathbb{E}_x f_k(x)||_2^2$$

$$\geq C\frac{\beta_k n_k}{2}\mathbb{E}_x||f_{k-1}(x)||_2 - \mathbb{E}_x\mathbb{E}_y\sum_{i=1}^{n_k}\mathbb{E}_{w_i}\phi(\langle w_i, f_{k-1}(x)\rangle)\phi(\langle w_i, f_{k-1}(y)\rangle)$$

$$= C\frac{\beta_k n_k}{2}\mathbb{E}_x||f_{k-1}(x)||_2 - n_k\mathbb{E}_x\mathbb{E}_y\mathbb{E}_{w_1}\phi(\langle w_1, f_{k-1}(x)\rangle)\phi(\langle w_1, f_{k-1}(y)\rangle)$$

$$\geq C\sqrt{n_0}\beta_k n_k \prod_{l=1}^{k-1}\sqrt{\beta_l}\sqrt{n_l} - n_k\beta_k$$

$$= C\sqrt{n_0}\beta_k n_k \prod_{l=1}^{k-1}\sqrt{\beta_l}\sqrt{n_l}$$

where to get the second inequality we have used lemma A.5, Jensen's inequality and the fact that $|\phi(x)| \leq 1$. In order to get an upper bound we observe

$$\mathbb{E}_{W_k}\mathbb{E}_x||f_k(x) - \mathbb{E}_x[f_k(x)]||_2^2 \leq \mathbb{E}_{W_k}\mathbb{E}_x||f_k(x)||_2^2$$

$$\leq \frac{C\beta_k n_k}{2}\mathbb{E}_x||f_{k-1}(x)||_2$$

$$\leq C\sqrt{n_0}\beta_k n_k \prod_{l=1}^{k-1}\sqrt{\beta_l}\sqrt{n_l}.$$

Applying Bernstein's inequality Vershynin (2018) we get

$$\frac{1}{2}\mathbb{E}_{W_k}\mathbb{E}_x||f_k(x) - \mathbb{E}_x[f_k(x)]||_2^2 \leq \mathbb{E}_x||f_k(x) - \mathbb{E}_x[f_k(x)]||_2^2 \leq \frac{3}{2}\mathbb{E}_{W_k}\mathbb{E}_x||f_k(x) - \mathbb{E}_x[f_k(x)]||_2^2$$

w.p. $\geq 1 - exp(-\Omega(n_k))$ over $W_k$. Taking the intersection of that event, together with the conditioned event over $(W_l)_{l=1}^{k-1}$ gives the statement of the lemma. $\square$

**Lemma A.8.** *For the activation function* $\phi(x) = e^{-x^2/s^2}$ *we have that*

$$||\Sigma_k(x)||_F^2 = \Theta\left(\frac{\beta_k^3}{s} n_k n_0^{3k/2} \prod_{l=1}^{k-1} \beta_l^3 n_l^3\right)$$

*w.p.* $\geq 1 - \sum_{l=1}^{k} 2exp(-\Omega(n_l)) - 2exp(-\Omega(\sqrt{n_0}))$.

*Proof.* We first observe that lemma A.4 implies that $||f_{k-1}(x)|| \neq 0$ w.p. $\geq 1 - \sum_{l=0}^{k-1} 2exp(-2s\Omega(n_k))$ which in turn implies that $f_{k-}(x) \neq 0$ w.h.p. $\geq 1 - \sum_{l=0}^{k-1} 2exp(-2s\Omega(n_k))$ over $(W_l)_{l=1}^{k-1}$ and $x$. We condition on that event and obtain bounds on $W_k$. Taking the intersection of the two events will then complete the proof.

Write $W_k = [w_1, \ldots, w_{n_k}]$. Then $||\Sigma_k(x)||_F^2 = \sum_{i=1}^{n_k} \phi'(\langle f_{k-1}(x), w_i \rangle)^2$. Thus $\mathbb{E}_{W_k}||\Sigma_k(x)||_F^2 = n_k \mathbb{E}_{w_1}[\phi'(\langle f_{k-1}(x), w_1 \rangle)^2]$, by independence. We have

$$\mathbb{E}_{W_k}||\Sigma_k(x)||_F^2 = n_k \mathbb{E}_{w_1}[\phi'(\langle f_{k-1}(x), w_1 \rangle)^2]$$
$$= \frac{4n_k}{s^4} \mathbb{E}_{w_1}[\langle f_{k-1}(x), w_1 \rangle^2 e^{-\frac{2\langle f_{k-1}(x), w_1 \rangle^2}{s^2}}].$$

Using the fact that $\langle f_{k-1}(x), w_1 \rangle$ is a univariate random variable distributed according to $\mathcal{N}(0, \beta_k^2 ||f_{k-1}(x)||_2^2)$. Thus the above expectation is equivalent to $\frac{4n_k}{s^4} \mathbb{E}_w[w^2 e^{-\frac{2w^2}{s^2}}]$ with $w \sim \mathcal{N}(0, \beta_k^2 ||f_{k-1}(x)||_2^2)$. We then compute

$$\mathbb{E}_w[w^2 e^{-\frac{2w^2}{s^2}}] = \int_{\mathbb{R}} w^2 e^{-\frac{2w^2}{s^2}} e^{-\frac{w^2}{\beta_k^2 ||f_{k-1}(x)||_2^2}} dw$$
$$= \int_{\mathbb{R}} w^2 e^{-\left(\frac{2}{s^2} + \frac{1}{\beta_k^2 ||f_{k-1}(x)||_2^2}\right)w^2} dw$$
$$= \int_{\mathbb{R}} w^2 e^{-\left(\frac{2\beta_k^2 ||f_{k-1}(x)||_2^2 + s^2}{\beta_k^2 s^2 ||f_{k-1}(x)||_2^2}\right)w^2} dw$$
$$= \frac{-1}{2}\left(\frac{\beta_k^2 s^2 ||f_{k-1}(x)||_2^2}{2\beta_k^2 ||f_{k-1}(x)||_2^2 + s^2}\right) \int_{\mathbb{R}} w \frac{d}{dw}\left(e^{-\left(\frac{2\beta_k^2 ||f_{k-1}(x)||_2^2 + s^2}{\beta_k^2 s^2 ||f_{k-1}(x)||_2^2}\right)w^2}\right) dw$$
$$= \frac{1}{2}\left(\frac{\beta_k^2 s^2 ||f_{k-1}(x)||_2^2}{2\beta_k^2 ||f_{k-1}(x)||_2^2 + s^2}\right) \int_{\mathbb{R}} e^{-\left(\frac{2\beta_k^2 ||f_{k-1}(x)||_2^2 + s^2}{\beta_k^2 s^2 ||f_{k-1}(x)||_2^2}\right)w^2} dw$$
$$= \frac{1}{2}\left(\frac{\beta_k^2 s^2 ||f_{k-1}(x)||_2^2}{2\beta_k^2 ||f_{k-1}(x)||_2^2 + s^2}\right)^{3/2}.$$

Thus we get

$$\mathbb{E}_{W_k}[||\Sigma_k(x)||_F^2] = \frac{4n_k}{2s^4}\left(\frac{\beta_k^2 s^2 ||f_{k-1}(x)||_2^2}{2\beta_k^2 ||f_{k-1}(x)||_2^2 + s^2}\right)^{3/2}$$

which implies

$$\mathbb{E}_{W_k}[||\Sigma_k(x)||_F^2] = \Theta\left(\frac{\beta_k^3}{s} n_k n_0^{3k/2} \prod_{l=1}^{k-1} \beta_l^3 n_l^3\right).$$

We now apply Hoeffding's inequality Vershynin (2018) to get

$$\left|||\Sigma_k(x)||_F^2 - \mathbb{E}_{W_k}||\Sigma_k(x)||_F^2\right| \leq \frac{1}{2}\mathbb{E}_{W_k}||\Sigma_k(x)||_F^2$$

*w.p.* $\geq 1 - 2exp(-\frac{\left(\mathbb{E}_{W_k}||\Sigma_k(x)||_F^2\right)^2}{4n_k})$. Using the estimate for $\mathbb{E}_{W_k}||\Sigma_k(x)||_F^2$ that we obtained and taking the intersection of the two events proves the lemma

$\square$

**Lemma A.9.** *Let* $\phi(x) = e^{-x^2/s^2}$. *Then*

$$\left\|\Sigma_k(x)\prod_{l=k+1}^{p} W_l\Sigma_l(x)\right\|_F^2 = \Theta\left(\frac{\beta_k^3 n_0^{3k/2}}{s^3}\left(\prod_{l=1}^{k-1}\beta_l^3 n_l^3\right)\left(\prod_{l=k}^{p}\beta_l n_l\right)\right)$$

*w.p.* $\geq 1 - \sum_{l=0}^{p} 2exp(-\Omega(n_l))$ *over* $(W_l)_{l=1}^{p}$ *and* $x \sim \mathcal{P}$.

*Proof.* We want to bound $||\Sigma_k(X)\prod_{l=k+1}^{p} W_l\Sigma_l(x)||_F^2$ for $k \leq p \leq l-1$, and any $k \in [L-1]$, $x \sim \mathcal{P}$.

When $p = k$, the quantity reduces to $||\Sigma_k(x)||_F^2$, which we know how to bound by lemma A.8.

Let $B(p) = \Sigma_k(x)\prod_{l=k+1}^{p} W_l\Sigma_l(x) = \Sigma_k(x)\left(\prod_{l=k+1}^{p-1} W_l\Sigma_l(x)\right)W_p\Sigma_p(x) = B(p-1)W_p\Sigma_p(x)$.

Write $W_p = [w_1, \ldots, w_{n_p}]$ and observe that

$$||B(p)||_F^2 = \sum_{i=1}^{n_p} ||B(p-1)w_i||_2^2 \phi'(\langle f_{p-1}(x), w_i\rangle)^2.$$

Taking the expectation we obtain

$$\mathbb{E}_{W_p}||B(p)||_F^2 = n_p\mathbb{E}_{w_1}||B(p-1)w_1||_2^2 \phi'(\langle f_{p-1}(x), w_i\rangle)^2.$$

The derivative $\phi'(\langle f_{p-1}(x), w_i\rangle)^2 = \frac{4}{s^2}\langle f_{p-1}(x), w_i\rangle^2 e^{\frac{-2\langle f_{p-1}(x), w_i\rangle^2}{s^2}}$.

Pick a piecewise non-negative, non-zero, measurable locally constant function $\chi$ so that

$0 \leq \chi(x) \leq \frac{4x^2}{s^2}e^{\left(\frac{-2x^2}{s^2}\right)}$. Then observe that

$$\mathbb{E}_{W_p}||B(p)||_F^2 \geq n_p\mathbb{E}_{w_1}||B(p-1)w_1||_2^2\chi s^2$$
$$= n_p s^2 \beta_p^2 ||B(p-1)||_F^2.$$

To get an upper bound, we simply observe that $\phi'$ is a bounded function. Therefore,

$$\mathbb{E}_{W_p}||B(p)||_F^2 \leq s^2 n_p \beta_p ||B(p-1)||_F^2.$$

By induction, applying lemma A.8, we get

$$\mathbb{E}_{W_p}||B(p)||_F^2 = \Theta\left(\frac{\beta_k^3 n_0^{3k/2}}{s^3}\left(\prod_{l=1}^{k-1}\beta_l^3 n_l^3\right)\left(\prod_{l=k}^{p}\beta_l n_l\right)\right).$$

Once we have an expectation bound we can apply Bernstein's inequality Vershynin (2018). In order to do this, we need to compute the sub-Gaussian norm. By using the fact that $\phi'(x)^2$ is a bounded function we have

$$\left\|||B(p-1)w_1||_2^2 \phi'(\langle f_{p-1}(x), w_i\rangle)^2\right\|_{\psi_1} \leq C\left\|||B(p-1)w_1||_2\right\|_{\psi_2}^2$$
$$\leq C\beta_p^2 ||B(p-1)||_F^2$$

for some $C > 0$.

Once we have the sub-Gaussian norm estimate, we can apply Bernstein's inequality Vershynin (2018) to get

$$\frac{1}{2}\mathbb{E}_{W_p}||B(p)||_F^2 \leq ||B(p)||_F^2 \leq \frac{3}{2}\mathbb{E}_{W_p}||B(p)||_F^2$$

w.p. $\geq 1 - 2exp(-\Omega(n_p))$ over $W_p$. Taking the intersection of this event with the previous events over $(W_l)_{l=1}^{p-1}$ and $x$ gives the result. $\qquad\square$

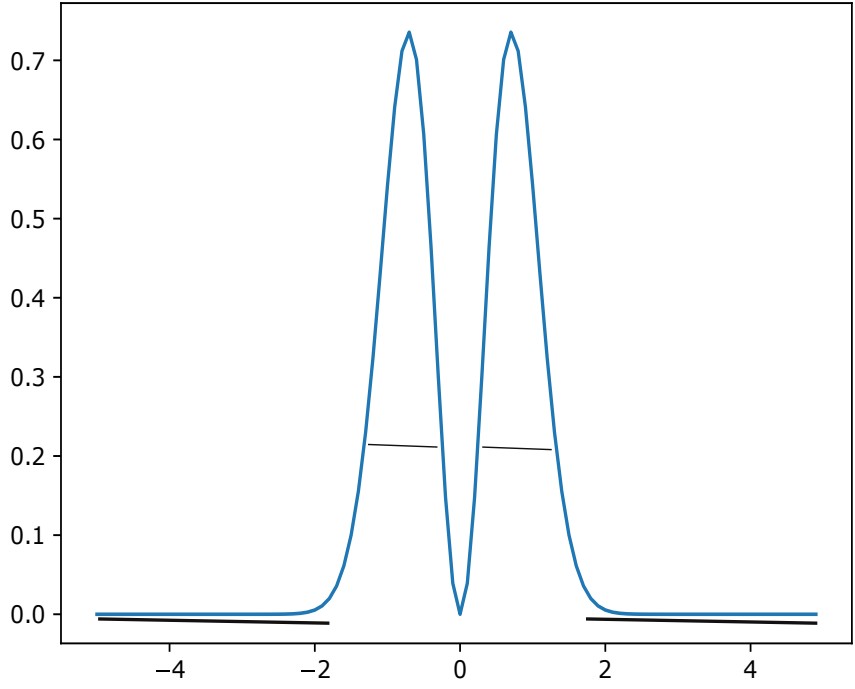

Figure 5: An example of a $\chi$, black curve.

### A.1.5 PROOF OF LEMMA A.2

**Lemma A.10.** *Let* $\phi(x) = e^{-x^2/s^2}$, *then* $||\Sigma_k(x)||_{op} \leq \frac{2}{s}$.

*Proof.* For the activation function $e^{-x^2/s^2}$, the derivative is $\frac{-2x}{s^2}e^{-x^2}$. The maximum of this derivative occurs at the point $\frac{-s}{\sqrt{2}}$ at which it has the value $\frac{\sqrt{2}}{s}e^{-1/2}$. The result follows. $\qquad\square$

**Lemma A.11.** *Let* $A = \Sigma_k(x) \prod_{l=k+1}^{L-1} W_l \Sigma_l(x)$ *and let* $\phi(x) = e^{-x^2/s^2}$. *Then*

$$||A||_{op}^2 = \mathcal{O}\left(\frac{n_k}{s\min_{l\in[k,L-1]} n_l} \prod_{l=k+1}^{L-1} n_l\beta_l^2\right)$$

*w.p.* $\geq 1 - \sum_{l=0}^k 2exp(-\Omega(n_l))$ *over* $(W_l)_{l=1}^k$ *and* $x$.

*Proof.* We first note that we have the estimate

$$||A||_{op} \leq ||\Sigma(x)||_{op}\left\|\prod_{l=k+1}^{L-1} W_l\Sigma_l(x)\right\|_{op}. \tag{35}$$

We then observe that $||\Sigma_k(x)||_{op}$ can be bounded by lemma A.10. This means we need only bound $\left\|\prod_{l=k+1}^{L-1} W_l\Sigma_l(x)\right\|_{op}$. The proof of this follows by induction on the length $(L-1) - (k+1) = L - k - 2$.

The base case follows by applying operator norm bounds of Gaussian matrices, see theorem 2.13 in Davidson & Szarek (2001).

$$\left\|W_{L-1}\Sigma_{L-1}(x)\right\|_{op} \leq C(s)\left\|W_{L-1}\right\|_{op}^2 = \mathcal{O}(\beta_{L-1}^2 \max\{n_{L-1}, n_{L-2}\}).$$

The general case now follows the $\epsilon$-net argument used in Nguyen et al. (2021).

$\square$

***Proof of lemma A.2.*** We need to estimate the quantity

$$\left|\left|\Sigma_k(x)\left(\prod_{l=k+1}^{L-1} W_l\Sigma_l(x)\right)W_L\right|\right|_2^2.$$

Let $A = \Sigma_k(x)\prod_{l=k+1}^{L-1} W_l\Sigma_l(x)$. By lemma A.9 we have that

$$||A||_F^2 = \Theta\left(\frac{\beta_k^3 n_0^{3k/2}}{s^3}\left(\prod_{l=1}^{k-1} \beta_l^3 n_l^3\right)\left(\prod_{l=k}^{L-1} \beta_l n_l\right)\right)$$

w.p. $\geq 1 - \sum_{l=0}^{p} 2exp(-\Omega(n_l))$ over $(W_l)_{l=1}^{p}$ and $x \sim \mathcal{P}$.

Lemma A.11 then gives the operator norm estimate

$$||A||_{op}^2 = \mathcal{O}\left(\frac{n_k}{s\min_{l\in[k,L-1]} n_l}\prod_{l=k+1}^{L-1} n_l\beta_l^2\right)$$

w.p. $\geq 1 - \sum_{l=0}^{k} 2exp(-\Omega(n_l))$ over $(W_l)_{l=1}^{k}$ and $x$.

As $A$ only depends on $(W_l)_{l=1}^{L-1}$ and $x$, we condition on the above two events over $(W_l)_{l=1}^{L-1}$ and $x$, and obtain a bound over $W_L$. Applying the Hanson-Wright inequality Vershynin (2018) we get

$$\left|||AW_L||_2^2 - \mathbb{E}_{W_L}||AW_L||_2^2\right| \leq \frac{3}{2}\mathbb{E}_{W_L}||AW_L||_2^2$$

w.p. $\geq 1 - exp\left(-\Omega\left(\frac{||A||_F^2}{\max_i ||(AW_L)_i||_{\psi_2}}\right)\right)$.

Note that $||(AW_L)_i||_2^2 \leq ||B||_{op}^2||W_L||_2^2$. It follows that for each $i$ that $||(AW_L)_i||_{\psi_2} = \mathcal{O}(||B||_{op})$. We therefore find that

$$||AW_L||_2^2 = \Theta\left(\frac{\beta_k^3 n_0^{3k/2}}{s^3}\left(\prod_{l=1}^{k-1} \beta_l^3 n_l^3\right)\left(\prod_{l=k}^{L} \beta_l n_l\right)\right)$$

w.p. $\geq 1 - 2exp(-\Omega(n_k))$. By taking the intersection of this event with the one we conditioned over, we get the result. $\square$

### A.1.6 PROOF OF THEOREM A.3

We start with the following lemma, whose proof is given in E.3 of (Ngyuen).

**Lemma A.12.** *Let $\widetilde{F}_k = F_k - \mathbb{E}_X[F_k]$ denote the centred features. Let $\mu = \mathbb{E}_{x\sim\mathcal{P}}[f_k(x)] \in \mathbb{R}^{n_k}$ and let $\Lambda = Diag(F_k\mu - ||\mu||_2^2 1_N)$, where $1_N \in \mathbb{R}^N$ is the column vector of $1's$. Then*

$$F_k F_k^T \geq \left(\widetilde{F}_k\widetilde{F}_k^T - \frac{\Lambda 1_N 1_N^T \Lambda}{||\mu||_2^2}\right)$$

*where $\geq$ sign is used in the sense of positive semi-definite matrices, meaning*

$$F_k F_k^T - \left(\widetilde{F}_k\widetilde{F}_k^T - \frac{\Lambda 1_N 1_N^T \Lambda}{||\mu||_2^2}\right) \geq 0.$$

***Proof of theorem A.3.*** By lemma A.12, in order to bound $\lambda_{\min}(F_k F_k^T)$ is suffices to bound $\lambda_{\min}(\widetilde{F}_k\widetilde{F}_k^T - \frac{\Lambda 1_N 1_N^T \Lambda}{||\mu||_2^2})$. The proof will focus on bounding this latter quantity.

By Weyl's inequality we have

$$\lambda_{\min}\left(\widetilde{F}_k\widetilde{F}_k^T - \frac{\Lambda 1_N 1_N^T \Lambda}{||\mu||_2^2}\right) \geq \lambda_{\min}(\widetilde{F}_k\widetilde{F}_k^T) - \lambda_{\max}(\frac{\Lambda 1_N 1_N^T \Lambda}{||\mu||_2^2})). \tag{36}$$

We start by bounding $\lambda_{\min}(\widetilde{F}_k \widetilde{F}_k^T)$.

By the Gershgorin circle theorem we have

$$\lambda_{\min}(\widetilde{F}_k \widetilde{F}_k^T) \geq \min_{i \in [N]} ||(\widetilde{F}_k)_{i:}||_2^2 - N max_{i \neq j} |\langle (\widetilde{F}_k)_{i:}, (\widetilde{F}_k)_{j:} \rangle| \tag{37}$$

$$\lambda_{\min}(\widetilde{F}_k \widetilde{F}_k^T) \leq \max_{i \in [N]} ||(\widetilde{F}_k)_{i:}||_2^2 + N max_{i \neq j} |\langle (\widetilde{F}_k)_{i:}, (\widetilde{F}_k)_{j:} \rangle|. \tag{38}$$

By lemma A.6, we have for all $i \in [N]$ that

$$||f_k(x_i) - \mathbb{E}_x[f_k(x)]||_2^2 = \Theta \left( \sqrt{n_0} \beta_k n_k \prod_{l=1}^{k-1} \sqrt{\beta_l} \sqrt{n_l} \right) \tag{39}$$

w.p. $\geq 1 - N exp \left( - \Omega \left( \frac{\min_{l \in [0,k]} n_l}{\prod_{l=1}^{k-1} log(n_l)} \right) \right) - \sum_{l=1}^k exp(-\Omega(n_l))$ over $(W_l)_{l=1}^k$ and $x$.

The goal is to find a bound for $|\langle (\widetilde{F}_k)_{i:}, (\widetilde{F}_k)_{j:} \rangle|$. By assumption A4 we have that

$$||f_k(x) - \mathbb{E}_x f_k(x)||_{Lip}^2 = \mathcal{O} \left( \frac{1}{s^k \min_{l \in [0,k]} n_l} \beta_k n_k \prod_{l=1}^{k-1} \sqrt{\beta_l} \sqrt{n_l} \prod_{l=1}^{k-1} Log(n_l) \right).$$

w.p. $\geq 1 - \sum_{l=1}^k 2exp(-\Omega(n_l))$ over $(W_l)_{l=1}^k$, where we used the fact that $f_k(x) - \mathbb{E}_x f_k(x)$ and $f_k(x)$ have the same Lipshitz constant.

We are going to condition on the intersection of the above event over $(W_l)_{l=1}^k$ and the event defined by equation 39 over $(W_l)_{l=1}^k$ and $x_j$ and derive bounds over $x_i$. Since we have conditioned on $x_j$, $|\langle (\widetilde{F}_k)_{i:}, (\widetilde{F}_k)_{j:} \rangle|$ is a function of $x_i$ for every $i \neq j$. We then have

$$\left| \left| |\langle (\widetilde{F}_k)_{i:}, (\widetilde{F}_k)_{j:} \rangle| \right| \right|_{Lip} \leq ||(\widetilde{F}_k)_{j:}||_2^2 ||f_k(x_i) - \mathbb{E}_x f_k(x_i)||_{Lip}^2$$

$$= \mathcal{O} \left( \frac{\sqrt{n_0}}{s^k \min_{l \in [0,k]} n_l} \beta_k^2 n_k^2 \prod_{l=1}^k \beta_l n_l \prod_{l=1}^{k-1} Log(n_l) \right)$$

using the above two asymptotic estimates we have conditioned on. Note that the above holds for all $x_i \neq x_j$.

Applying our concentration assumption A3, and taking the union of the above estimate over all $x_i \neq x_j$ we have

$$\mathbb{P} \left( \max_{i \in [N], i \neq j} |\langle (\widetilde{F}_k)_{i:}, (\widetilde{F}_k)_{j:} \rangle| \geq t \right) \leq (N-1) exp \left( - \frac{t^2}{\mathcal{O} \left( \frac{\sqrt{n_0}}{s^k \min_{l \in [0,k]} n_l} \beta_k^2 n_k^2 \prod_{l=1}^k \beta_l n_l \prod_{l=1}^{k-1} Log(n_l) \right)} \right).$$

Choosing $t = N^{-1} \sqrt{n_0} \beta_k n_k \prod_{l=1}^{k-1} \sqrt{\beta_l} \sqrt{n_l}$. We have

$$N max_{i \neq j} |\langle (\widetilde{F}_k)_{i:}, (\widetilde{F}_k)_{j:} \rangle| \leq \beta_k n_k \prod_{l=1}^{k-1} \sqrt{\beta_l} \sqrt{n_l}$$

w.p. $\geq 1 - (N-1) exp \left( -\Omega \left( \frac{\min_{l \in [0,k]} n_l}{s^k \prod_{l=1}^{k-1} Log(n_l)} \right) \right) - \sum_{l=1}^k 2exp(-\Omega(n_l))$. Therefore we can take a union of the bounds for each $j \in [N]$ to obtain

$$N max_{i \neq j} |\langle (\widetilde{F}_k)_{i:}, (\widetilde{F}_k)_{j:} \rangle| = o \left( \beta_k n_k \prod_{l=1}^{k-1} \sqrt{\beta_l} \sqrt{n_l} \right)$$

w.p. $\geq 1 - N(N-1) exp \left( -\Omega \left( \frac{\min_{l \in [0,k]} n_l}{s^k \prod_{l=1}^{k-1} Log(n_l)} \right) \right) - N \sum_{l=1}^k 2exp(-\Omega(n_l))$.

We then obtain that

$$\lambda_{\min}(\widetilde{F_k}\widetilde{F_k}^T) = \Theta\left(\beta_k n_k \prod_{l=1}^{k-1} \sqrt{\beta_l}\sqrt{n_l}\right) \tag{40}$$

w.p. $\geq 1 - N(N-1)exp\left(-\Omega\left(\frac{s^k \min_{l\in[0,k]} n_l}{\prod_{l=1}^{k-1} Log(n_l)}\right)\right) - N\sum_{l=1}^{k} 2exp(-\Omega(n_l))$. This bounds the first term on the right hand side of equation 36. We move on to bounding the second term on the right hand side of equation 36.

We want to bound the maximum eigenvalue of the quantity $\frac{\Lambda 1_N 1_N^T \Lambda}{||\mu||_2^2}$. The maximum eigenvalue is the operator norm, therefore we will obtain an estimate for the operator norm. As a start we have the simple estimate

$$\left|\left|\frac{\Lambda 1_N 1_N^T \Lambda}{||\mu||_2^2}\right|\right|_{op} \leq \left|\left|\frac{\Lambda}{||\mu||_2}\right|\right|_{op}^2.$$

We define an auxiliary random variable $g : \mathbb{R}^d \to \mathbb{R}$ by $g(x) = \langle f_k(x), \mu \rangle$. Note that $\Lambda_{ii} = g(x_i) - \mathbb{E}_x[g(x)]$ and that $||g||_{Lip}^2 \leq ||\mu||_2^2 ||f_k||_{Lip}^2$. Therefore, applying Liptshitz concentration, we get

$$\mathbb{P}\left(|\Lambda_{ii}| \geq t\right) \leq exp\left(-\frac{t^2}{2||\mu||_2^2 ||f_k||_{Lip}^2}\right).$$

From lemma A.5, we have

$$||\mu||_2^2 = \Theta\left(\sqrt{n_0}\beta_k n_k \prod_{l=1}^{k-1} \sqrt{\beta_l}\sqrt{n_l}\right). \tag{41}$$

Furthermore, our assumption on the Lipshitz constant (A4) gives the estimate

$$||f_k(x)||_{Lip}^2 = \mathcal{O}\left(\frac{1}{s^k \min_{l\in[0,k]} n_l}\beta_k n_k \prod_{l=1}^{k-1} \sqrt{\beta_l}\sqrt{n_l} \prod_{l=1}^{k-1} Log(n_l)\right). \tag{42}$$

If we take $t = \frac{1}{N}||\mu||_2^2$, and take a union bound over all the samples $\{x_i\}$ and the events defined by equation 41, equation 42, we get the estimate

$$\left|\left|\frac{\Lambda}{||\mu||_2}\right|\right|_{op}^2 = \mathcal{O}\left(\frac{1}{N^2}\sqrt{n_0}\beta_k n_k \prod_{l=1}^{k-1} \sqrt{\beta_l}\sqrt{n_l}\right) \tag{43}$$

w.p. $\geq 1 - Nexp\left(-\Omega\left(\frac{\min_{l\in[0,k]} n_l}{s^k N^2 \prod_{l=1}^{k-1} Log(n_l)}\right)\right) - \sum_{l=1}^{k} 2exp(-\Omega(n_l))$.

Putting the estimate for $\lambda_{\min}\left(\widetilde{F_k}\widetilde{F_k}^T\right)$ and $\left|\left|\frac{\Lambda 1_N 1_N^T \Lambda}{||\mu||_2^2}\right|\right|_{op}$ together we obtain

$$\lambda_{\min}\left(F_k F_k^T\right) = \Theta\left(\sqrt{n_0}\beta_k n_k \prod_{l=1}^{k-1} \sqrt{\beta_l}\sqrt{n_l}\right)$$

w.p. $\geq 1 - N(N-1)exp\left(-\Omega\left(\frac{\min_{l\in[0,k]} n_l}{s^k N^2 \prod_{l=1}^{k-1} Log(n_l)}\right)\right) - N\sum_{l=1}^{k} 2exp(-\Omega(n_l))$. This completes the proof.

□

## A.2 EMPIRICAL RESULTS ON THE NTK AND ASSUMPTION 4

In this section we would like to show empirically that Assumption 4 is an an extremely mild assumption and further test the statement of thm.A.1, analogous to fig. 1.

The Lipschitz constant of the $k$-layer function $f_k$ can be expressed as the supremum, over each point in data space, of the operator norm of the Jacobian matrix as

$$||f_k||_{Lip} = \sup_{x\in\mathbb{R}^{n_0}} ||J(f_k)(x)||_{op}.$$

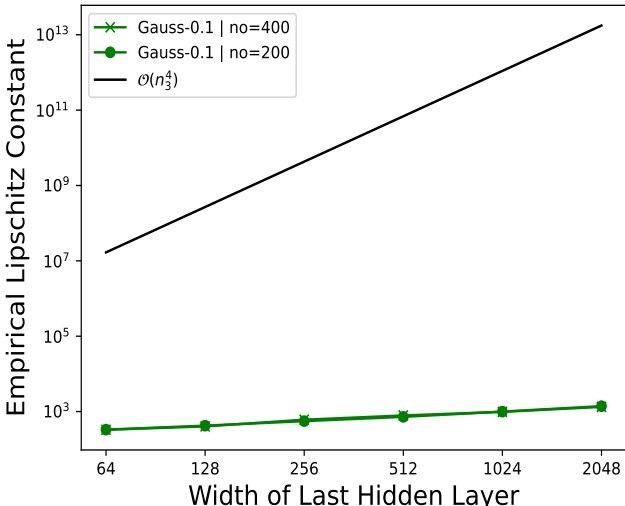

Figure 6: The empirical Lipshitz constant of a Gaussian-activated network over 1000 data points, where $n_1 = n_2 = 64$, $n_4 = 1$ and $n_3$ varying from 64 to 2048, when $n_0 = 200$ and $n_0 = 400$. This plot empirically confirms the assumption A4.

The exact computation of the Lipschitz constant of a deep network is considered as an NP-hard problem Virmaux & Scaman (2018). Therefore, for our experiment, we will consider the empirical Lipschitz constant. We obtain a sampled data set $X$, sampled from a fixed distribution $\mathcal{P}$; see sec. A.1.2. We then define the empirical Lipschitz constant of $f_k$ as

$$||f_k||_{ELip} = \max_{x \in X} ||J(f_k)(x)||_{op}.$$

Note that the empirical Lipschitz constant of $f_k$ is a lower bound for the true Lipschitz constant of $f_k$.

We computed the empirical Lipschitz constant of a $4-$layer Gaussian-activated network with variance $s^2 = 0.1^2$, $f_3$ over 1000 data points, drawn from a Gaussian distribution $\mathcal{N}(0, 1)$. The widths of the layers were fixed as $n_1 = n_2 = 64$, $n_4 = 1$, and $n_3$ was varied from 64 to 2048. We considered two different data dimensions, namely $n_0 = 200$ and $n_0 = 400$. Fig. 6 clearly shows that the empirical Lipschitz constant grows much slower with width than a term that grows $\mathcal{O}(n_3^{1/2})$. This empirically supports the assumption A4 and demonstrates that the bound given in assumption A4 is an extremely loose bound.

## A.3 EQUILIBRATED PRECONDITIONING

Preconditioning is a fundamental topic in numerical solutions for linear systems of equations, typically represented as $Ax = b$. It involves the adjustment of the matrix $A$ to reduce its condition number. Linear systems with lower condition numbers are not only solved more accurately but also more efficiently. The conventional approach to preconditioning transforms the original system into a simpler one, denoted as $PAx = Pb$. The key challenge in preconditioning lies in selecting the appropriate multiplier $P$ that effectively reduces the initially high condition number, $cond(A)$, to a significantly smaller value, $cond(PA)$.

For this work we will primarily focus on diagonal preconditioners. These are preconditoners $P$ which are diagonal matrices. We will primarily use the

1. Jacobi preconditioners; For a given matrix $A$ the Jacobi preconditioner of $A$ is the matrix $P = diag(A)^{-1}$.
2. Row Equilibrated preconditioner; For a given matrix $A$ the Row equilibrated preconditioner associated to $A$ is the diagonal matrix $P = diag(||A_{i;}||)^{-1}$ whose diagonal terms are the inverse of the norms of the rows of $A$.

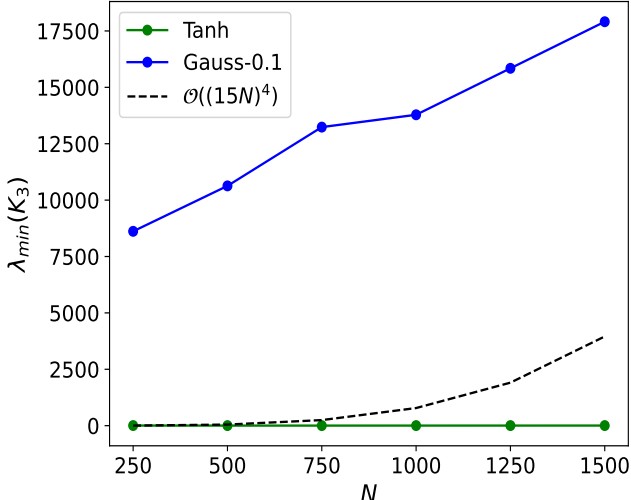

Figure 7: The minimum eigenvalue of the empirical NTK, where $n_0 = 400$, $n_1 = 15N$, and $n_2 = 400$. As predicted by Thm. A.1, $\lambda_{min}(K_L)$ for a Gaussian-activated network grows much faster than a Tanh-activated network and a quartic complexity of the width.

Analogous to row equilibration is column equilibration which generally performs the same. In this paper, we will primarily focus on row equilibration and often simply call it equilibration.

We recall that the main reason for focusing on the diagonal preconditioners given by row equilibration comes from Van Der Sluis' theorem.

**Theorem A.13** (Van der Sluis (1969)). *Let $A$ be a $n \times n$ matrix, $P$ an arbitrary diagonal matrix and $E$ the row equilibrated matrix built from $A$. Then $\kappa(EA) \leq \kappa(PA)$.*

Thm. 4.1 shows that out of the diagonal preconditioners, row equilibration is the optimal one for condition number reduction. A similar result also holds for column equilibration, see Van der Sluis (1969).

We now prove some results that shows why applying row equilibration is a suitable choice for reducing the condition number of a matrix. All proofs can be found in the appendix.

In Guggenheimer et al. (1995) the following estimate on the condition number of a non-singular $n \times n$ matrix $A$ is proven:

$$\kappa(A) \leq \frac{2}{|det(A)|}\left(\frac{||A||_F^2}{n}\right)^{\frac{n}{2}} := U(A) \tag{44}$$

The following lemma shows that equilibrating a non-singular matrix reduces the upper bound $U(A)$ by a precise factor that depends on $A$.

**Lemma A.14.** *Let $A$ be a non-singular matrix. Then equilibrating $A$ reduces the upper bound $U(A)$ by a factor of*

$$\frac{||A_{1;}|| \cdots ||A_{n;}||}{\left(\frac{||A||_F^2}{n}\right)^{n/2}}. \tag{45}$$

We note that the quantity in equation 45 is less than 1. While lem. A.14 shows that equilibrating a matrix can reduce an upper bound of its condition number it doesn't imply that the condition number actually decreases. Therefore, it is reasonable to study lower bounds. The following lemma provides a lower bound on the condition number.

**Lemma A.15.** *Let $A \in \mathbb{R}^{2xn}$ be a matrix with two rows and let the rows be denoted by $X_1$ and $X_2$. Let the angle between the two vectors $X_1$ and $X_2$ in $\mathbb{R}^n$ be denoted by $\theta(X_1, X_2)$. Suppose that*

$1 - cos(\theta(X_1, X_2)) = \epsilon$ *for some* $\epsilon \in (0, 1)$. *Then*

$$\kappa(A) \geq \left( \frac{1}{4\epsilon(2 - \epsilon)} \left( \frac{||X_1||}{||X_2||} + \frac{||X_2||}{||X_1||} + 2 \right) \right)^{\frac{1}{2}} \quad (46)$$

Lem. A.15 implies that if an $n \times n$ matrix $A$ has two rows that have angle $cos(\theta) = 1 - \epsilon$ for $\epsilon \in (0, 1)$. Then the condition number of $A$ is bounded below by $C\epsilon^{-1/2}$, for a constant $C$ that depends on the norms of the two rows.

**Lemma A.16.** *Let $A$ be a $n \times n$ matrix and assume $A$ has two rows $X_i$ and $X_j$ such that $cos(\theta(X_i, X_j)) = 1 - \epsilon$ for $\epsilon \in (0, 1)$. Let the lower bound on $\kappa(A)$ from lem. A.15 be denoted by $C_A\epsilon^{-1/2}$. Let $PA$ denote the row equilibration of $A$ and let the lower bound on $\kappa(PA)$, given by lem. A.15, be denoted by $C_{PA}\epsilon^{-1/2}$. Then $C_{PA} \leq C_A$.*

Lem. A.15-A.16 demonstrates that the process of equilibrating a matrix effectively lowers both the upper and lower bounds on its condition number. This compelling insight provides a strong rationale for considering equilibration as an effective strategy for lowering the condition number of a matrix.

***Proof of lem. A.14.*** Let $PA$ denote the equilibration of $A$. By equation 44, we have

$$\kappa(PA) \leq \frac{2}{|det(PA)|} \left( \frac{||PA||_F^2}{n} \right)^{\frac{n}{2}}. \quad (47)$$

Observe that the norm of each row of $PA$ is 1, yielding that $||PA||_F = n$. Hence $\frac{||PA||_F^2}{n} = 1$. Furthermore, by definition of $P$ we have that

$$det(P^{-1}) = ||A_{1;}|| \cdots ||A_{n;}|| \quad (48)$$

where $A_{i;}$ denotes the ith-row of $A$. Applying the arithmetic-geometric mean inequality we find

$$det(P^{-1}) \leq \left( \frac{(||A_{1;}|| + \cdots + ||A_{n;}||)^2}{n^2} \right)^{\frac{n}{2}}. \quad (49)$$

Using the fact that $(||A_{1;}|| + \cdots + ||A_{n;}||)^2 \leq n \cdot (||A_{1;}||^2 + \cdots + ||A_{n;}||^2)$, we find that

$$det(P^{-1}) \leq \left( \frac{||A_{1;}||^2 + \cdots + ||A_{n'}||^2}{n} \right)^{\frac{n}{2}}. \quad (50)$$

Then using the fact that $det(PA) = det(P)det(A)$ the result follows. $\square$

***Proof of lem. A.15.*** We consider the matrix $B = A^T A$ and note that the singular values of $A$ are precisely the square roots of the eigenvalues of $B$. Since $B$ is a $2 \times 2$ matrix a closed form expression for the eigenvalues $\lambda_1 \geq \lambda_2$ can be given by

$$\lambda_1 = \frac{X_1 X_1^T + X_2 X_2^T + ((X_1 X_1^T + X_2 X_2^T)^2 + (4X_1 X_2^T)^2)^{1/2}}{2}$$

$$\lambda_2 = \frac{X_1 X_1^T + X_2 X_2^T + ((X_1 X_1^T + X_2 X_2^T)^2 - (4X_1 X_2^T)^2)^{1/2}}{2}.$$

It follows that

$$\frac{\lambda_1}{\lambda_2} = \frac{[X_1 X_1^T + X_2 X_2^T + ((X_1 X_1^T + X_2 X_2^T)^2 + (4X_1 X_2^T)^2)^{1/2}]^2}{4(X_1 X_1^T X_2 X_2^T - (X_1 X_2^T)^2)}. \quad (51)$$

By assumption we have that $cos(\theta(X_1, X)2)) = 1 - \epsilon$, giving $X_1 X_2^T = (1 - \epsilon)^2 X_1 X_1^T X_2 X_2^T$. This implies

$$\frac{\lambda_1}{\lambda_2} \geq \frac{(X_1 X_1^T + X_2 X_2^T)^2}{4((1 - (1 - \epsilon)^2) X_1 X_1^T X_2 X_2^T} \quad (52)$$

which proves the result since $\kappa(A)$ is given by the square root of $\frac{\lambda_1}{\lambda_2}$. $\square$

We now given the proof of lem. A.16. To do so we will need a string of lemmas.

**Lemma A.17.** *Given an $n \times n$ matrix $A$ if $A$ has two rows $X_i$ and $X_j$ satisfying the assumptions of lem. A.15. Then*

$$\kappa(A) \geq \left( \frac{1}{4\epsilon(2-\epsilon)} \left( \frac{||X_i||}{||X_j||} + \frac{||X_i||}{||X_j||} + 2 \right) \right)^{\frac{1}{2}} := L(i,j). \tag{53}$$

*Proof.* The proof of this lemma follows easily from lem. A.15. □

**Lemma A.18.** *Let $A$ be a non-singular matrix $n \times n$, with $n > 0$, and let $\Theta$ denote the matrix whose entries are given by $cos(\theta_{ij})$, where $\theta_{ij}$ denotes the angle between the ith and jth row of $A$. Assume for $i \neq j$ that $\Theta_{ij} = 1 - \epsilon_{ij}$ for $\epsilon_{ij} \in (0,1)$. Let $\epsilon = \min_{i \neq j} \Theta_{ij} = cos(\theta_{IJ})$, where $I$ and $J$ denote the Ith and Jth rows of $A$, which we denote by $X_I$ and $X_J$ respectively. Then*

$$\kappa(A) \geq \frac{2}{n(n-1)} \cdot \left( \frac{1}{4\epsilon(2-\epsilon)} \left( \frac{||X_I||}{||X_J||} + \frac{||X_I||}{||X_J||} + 2 \right) \right)^{\frac{1}{2}} = L(I,J) := L(A). \tag{54}$$

*Proof.* The proof of this lemma follows by observing that by lem. A.17, $\kappa(A)$ can be lower bounded by each $L(i,j)$ (see lem. A.17 for the notation $L(i,j)$) for $i \neq j$. There are in total $\frac{(n-1)n}{2}$ of the $L(i,j)$. It therefore follows that

$$\frac{(n-1)n}{2}\kappa(A) \geq \sum_{i \neq j} L(i,j) \tag{55}$$

which implies

$$\kappa(A) \geq \frac{2}{n(n-1)} L(I,J). \tag{56}$$

□

**Theorem A.19.** *Given a non-singular $n \times n$ matrix $A$ with $n > 0$. We have that the equilibrated matrix $PA$ has $L(PA) \leq L(A)$. In particular, equilibrating a matrix lowers the upper bound $U(A)$ on $\kappa(A)$ and lowers the lower bound $L(A)$ on $\kappa(A)$.*

*Proof.* Consider the function $f : \mathbb{R}^n \times \mathbb{R}^n - \{(0,0)\} \to \mathbb{R}$ defined by

$$f(X,Y) = \frac{||X||}{||Y||} + \frac{||Y||}{||X||} + 2. \tag{57}$$

By computing the gradient of this function and solving for minima we see that the minimum of $f$ occurs at the points given by $\{X = Y\}$ and $\{X = -Y\}$. In particular the minimum of $f$ occurs on the points $(X,Y)$ such that $||X|| = ||Y||$ and the minimum value is exactly $4$. It follows that $L(i,j)_{PA} \leq L(i,j)_A$ for any $i \neq j$. The result follows. □

The proof of lem. A.16 then immediately follows from lem. A.19.

## A.4 EXPERIMENTS

### A.4.1 HARDWARE AND SOFTWARE

All experiments were run on a Nvidia RTX A6000 GPU. Furthermore, all the exerpiments were coded in PyTorch version 2.0.1.

**Experimental trials:** All experiments were each run 5 times and the average of the train, test and times are what is recorded.

**Hyperparameters:** The Gaussian and sine activation both have a hyperparamter given by the variance $s^2$ for the Gaussian and $f$, the frequency for sine. In general there is no principled way to choose these hyperparameters. Therefore, we ran several trials and found that for a Gaussian, $s = 0.1 - 0.4$ worked best. For a sine we found that a frequency of $1 - 10$ worked best.

## A.5 BURGERS' EQUATION

All networks used to test on this equation were 3 hidden layers with 128 neurons. We used synthetic data from:

```
https://github.com/jdtoscano94/Learning-Python-Physics-\
Informed-Machine-Learning-PINNs-DeepONets
```

In total we used 100 boundary and intial points which were randomly sampled using a uniform distribution from a boundary training set of 456 points. We used 10000 sampled PDE data points using a latin hypercube sampling strategy. The Gaussian networked used a variance of $s^2 = 0.1^2$ and the sine network used a frequency of $f = 10$.

All networks were trained with Adam on full batch and used the standard configuration in the adam paper Kingma & Ba (2014), with a learning rate of 1e-4. We note that some practitioners use much smaller learning rates but we found for Tanh networks these were proving unstable and leading to NaN results for the loss. In order to keep each experiment fair, we adopted 1e-4 as it worked well with all networks. All networks were trained for 15000 epochs on full batches.

## A.6 NAVIER-STOKES

For the Navier-Stokes example from sec. 5.2, we used a 3 hidden layer network with 128 neurons in each layer for each network. The data was synthetic data obtained from the github of the original paper Raissi et al. (2019): `https://github.com/maziarraissi/PINNs`

We used 5000 uniformly sampled points from a total training set of 1000000 points. All networks were trained with Adam with a learning rate of 1e-4 for 30000 epochs on full batches. For the Gaussian network we found that a variance of $s^2 = 0.4^2$ worked best and for the sine network we found a frequency of 2 worked best.

The training curves of all the loss functions are shown in fig. 8. The Gaussian-activated PINN performs the best overall, showing faster convergence than all other 3. Note that in Raissi et al. (2019) a Tanh network was used to reconstruct the velocity and pressure of the Navier-Stokes equations and in that case convergence only occurred at 80000 epochs.

## A.7 DIFFUSION EQUATION

We trained 6 different networks to fit the diffusion equation, see sec. 5.3. The networks were, a standard Tanh-PINN Raissi et al. (2019), Locally adaptive activation PINN (L-LAAF-PINN), with a Gaussian activation, Jagtap et al. (2020), Random Fourier Feature PINN (RFF-PINN) Wang et al. (2021), A PINNsformer Zhao et al. (2023), a Gaussian-activated PINN (G-PINN) and an Equilibrated Gaussian-activated PINN (EG-PINN).

G-PINN and EG-PINN employed a Gaussian with variance $s^2 = 0.2^2$.

Sec. 5.3 gave the training/testing results showing that EG-PINN is superior over all other networks.

Fig. 9 shows the training/testing errors of EG-PINN against a RFF-PINN and a stock standard Tanh-PINN. While the Tanh-PINN seems to start to converge around 17000 epochs in, the RFF-PINN struggles. EG-PINN on the other hand starts to immediately converge soon after a few thousand epochs.

Fig. 10 shows the training/testing errors of EG-PINN against a PINNsformer. The training/testing erros for the PINNsformer are extremely noisy and it was not showing any signs of convergence.

Fig. 11 shows the reconstruction of the RFF and PINNsformer networks. Both networks have struggled to find a solution to the PDE.

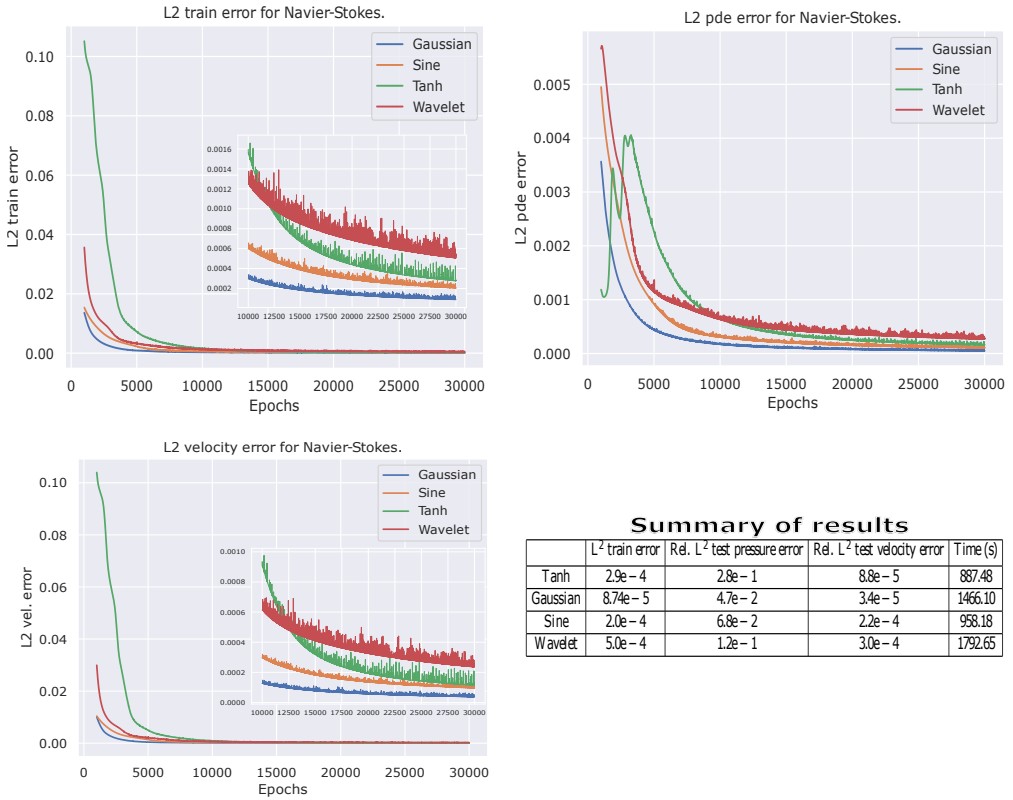

Figure 8: Training error of all loss functions. The Gaussian PINN converges much faster than all the other 3.

## A.8 POISSON'S EQUATION

We consider Poisson's Equation given by

$$-\Delta u = 25\pi^2 sin(5\pi) \text{ for } x \in [-1, 1] \tag{58}$$
$$u(-1) = u(1) = 0. \tag{59}$$

This PDE has a closed form solution given by $u(x) = sin(5\pi x)$.

We trained the same network frameworks from sec. 5.3, however all networks had two hidden layers with 128 neurons, except for PINNsformer which had 9 hidden layers and 128 neurons in each. We then how each does in reconstructing the solution to the above PDE. The above PDE is a low frequency PDE and standard PINN architectures do not have trouble finding an accurate solution. However, we will only trained each architecture for 5000 epochs thereby comparing whether each architecture can find a solution in a short number of iterations. We used 1000 PDE sample points and trained each PINN with the Adam optimizer, learning rate of 1e-4, on full batch.

Fig. 12, shows the train/test errors for the three Gaussian-activated PINNs, G-PINN, EG-PINN and L-LAAF-PINN. From the figure we see that EG-PINN has converged the fastest. Table 4 shows the training/testing results for each PINN. The first three PINNs employ a Gaussian activation function and achieve significantly better results than the other three. Furthermore, amongst the top three, EG-PINN performs significantly better achieving at least $10^3$ lower relative test accuracy.

Figs. 13-14 show the train/test errors for an EG-PINN against a Tanh-PINN and a PINNsformer. IN both cases the EG-PINN is able to converge in extremely less iterations while the Tanh-PINN and the PINNsformer struggle.

You may include other additional sections here.

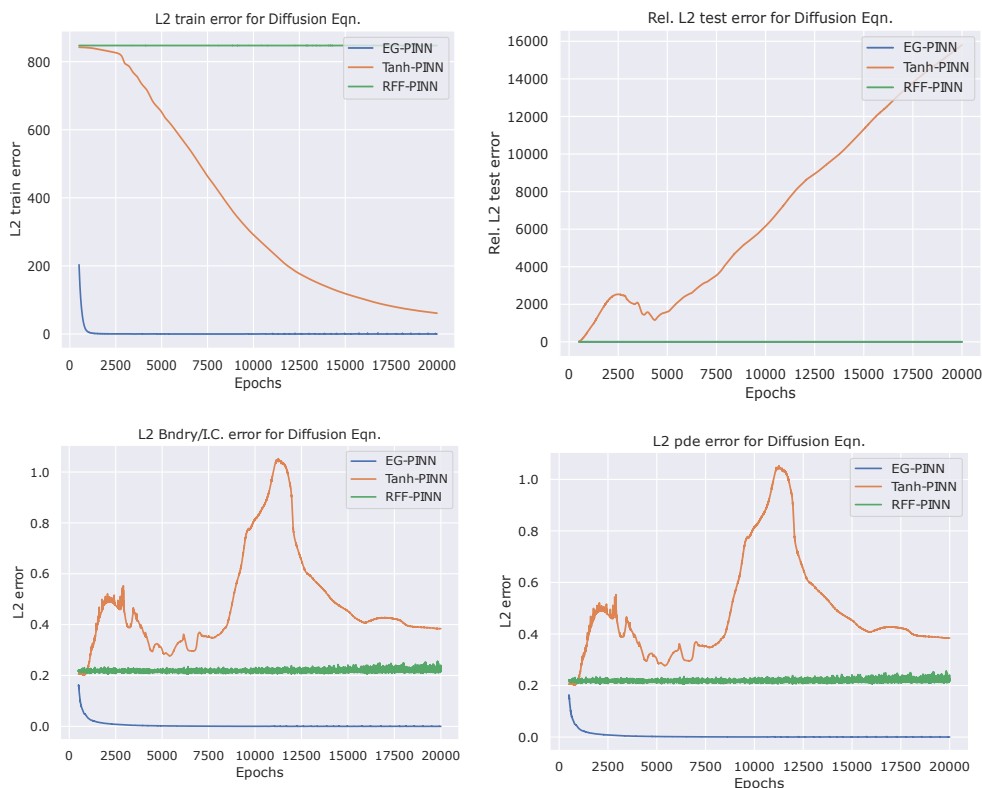

Figure 9: Train/test error on the Diffusion equation. EG-PINN converges extremely well while the other two struggle with all losses.

| | Train error | Rel. test error | Bndry condition error | Pde error | Time (s) |
|---|---|---|---|---|---|
| G-PINN | $1.3e-2$ | $1.1e-2$ | $4.7e-3$ | $8.5e-3$ | 20.01 |
| EG-PINN | $6.0e-4$ | $4.9e-7$ | $8.5e-7$ | $6.1e-4$ | 98.95 |
| L-LAAF-PINN | $7.9e-3$ | $9.0e-4$ | $1.3e-3$ | $6.6e-3$ | 20.22 |
| PINNsformer | $1.4e2$ | $2.4e0$ | $6.0e-4$ | $1.4e2$ | 267 |
| RFF-PINN | $3.0e4$ | $1.0e0$ | $1.8e-3$ | $3.1e4$ | 39.25 |
| Tanh-PINN | $9.3e2$ | $4.8e1$ | $3.1e1$ | $9.0e2$ | 13.78 |

Table 4: Training/Testing results for Poisson's equation. For each of the accuracy measures, EG-PINN achieves at least 10 times lower loss value when compared to many of the others.

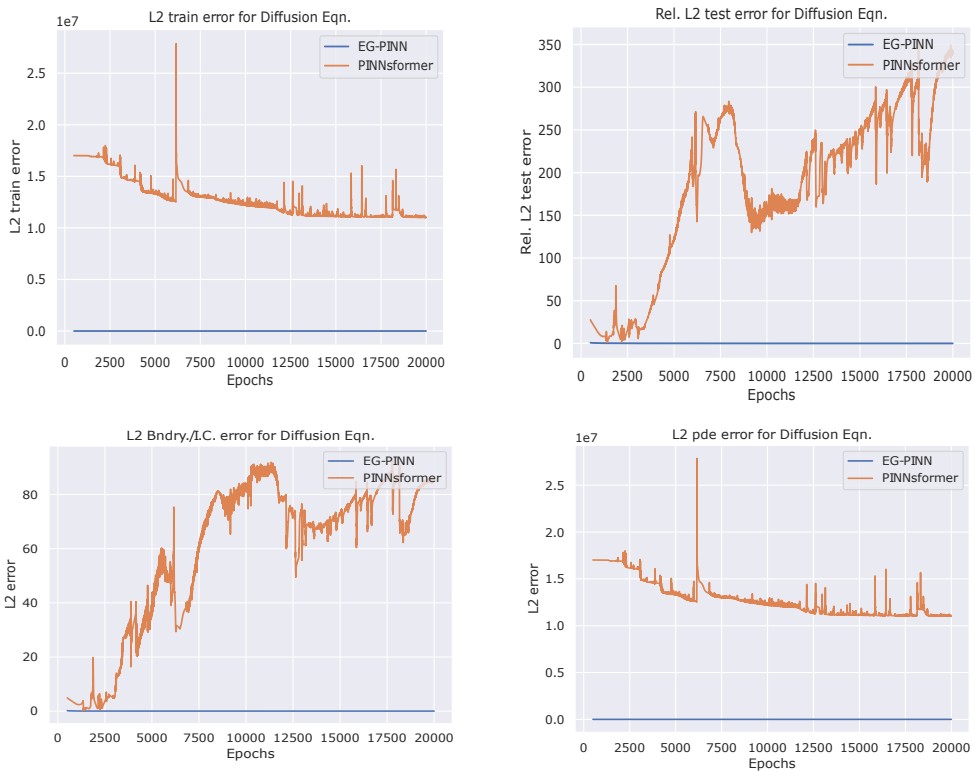

Figure 10: Train/test error on the Diffusion equation for an EG-PINN and a PINNsformer.

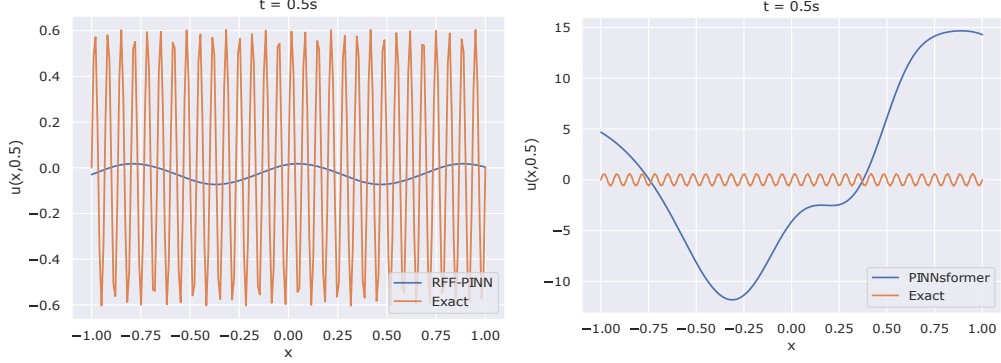

Figure 11: Reconstruction of an RFF-PINN and a PINNsformer on the diffusion equation. Both networks are unable to find the right solution.

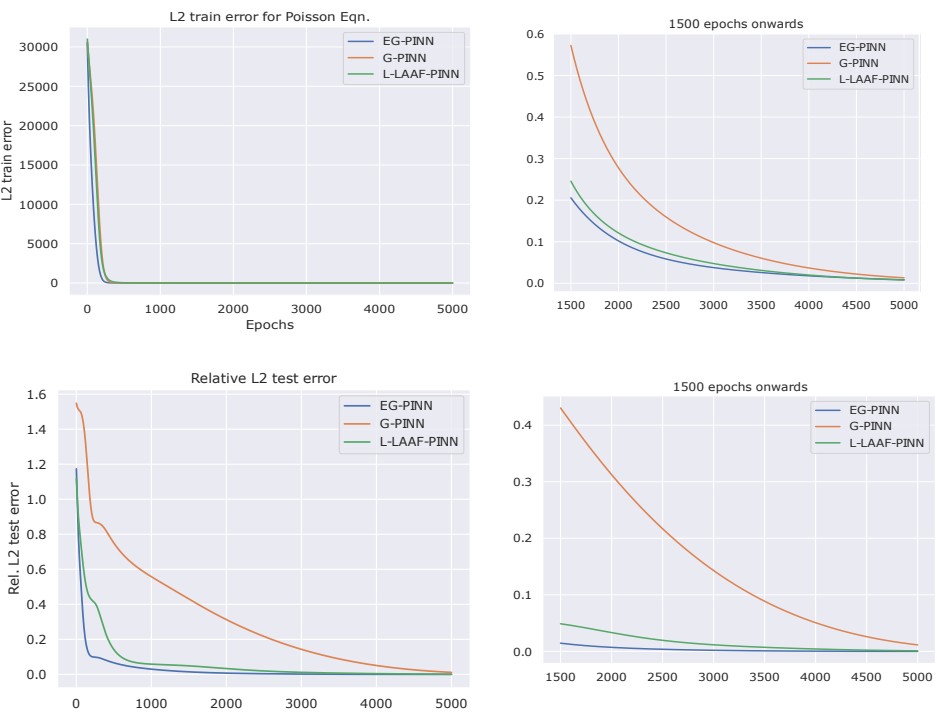

Figure 12: Train/test error for 3 Gaussian-activated networks on solving the Poisson problem. EG-PINN converges the fastest outperforming all other networks.

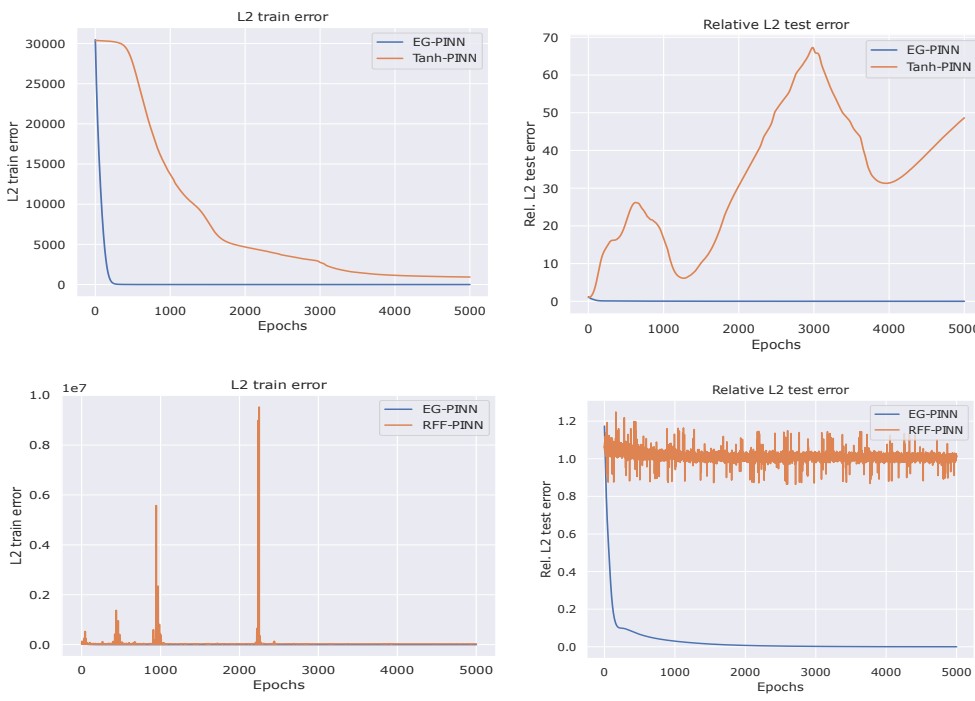

Figure 13: Train/test error for an EG-PINN vs a Tanh-PINN. The EG-PINN outperforms the Tanh-PINN dramatically.

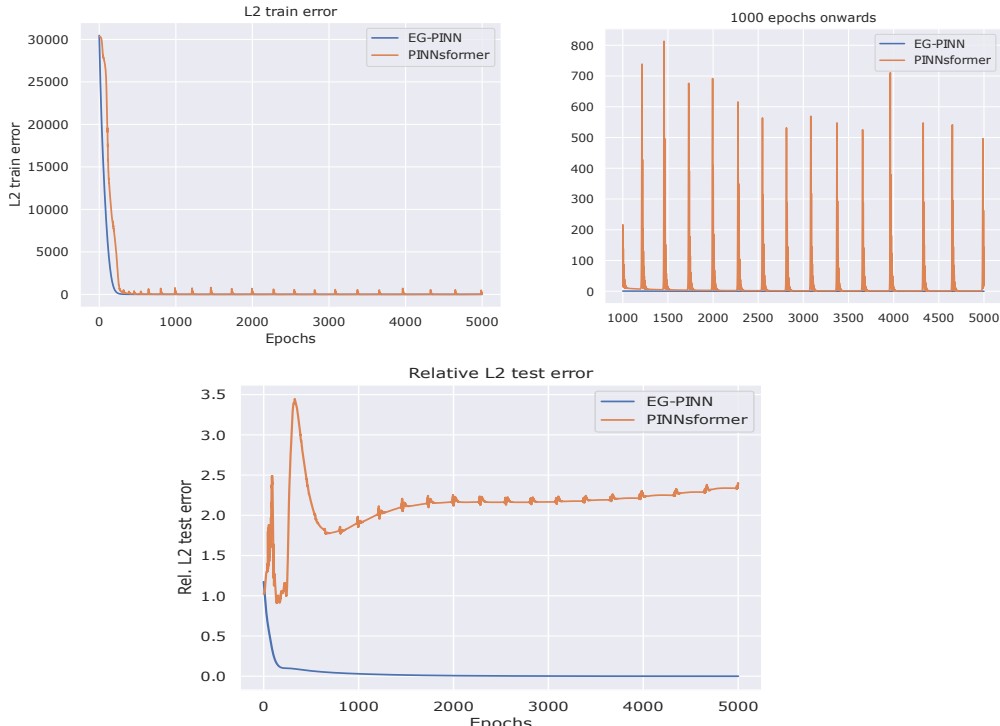

Figure 14: Train/test error for an EG-PINN vs a PINNsformer. The EG-PINN outperforms the PINNsformer dramatically.

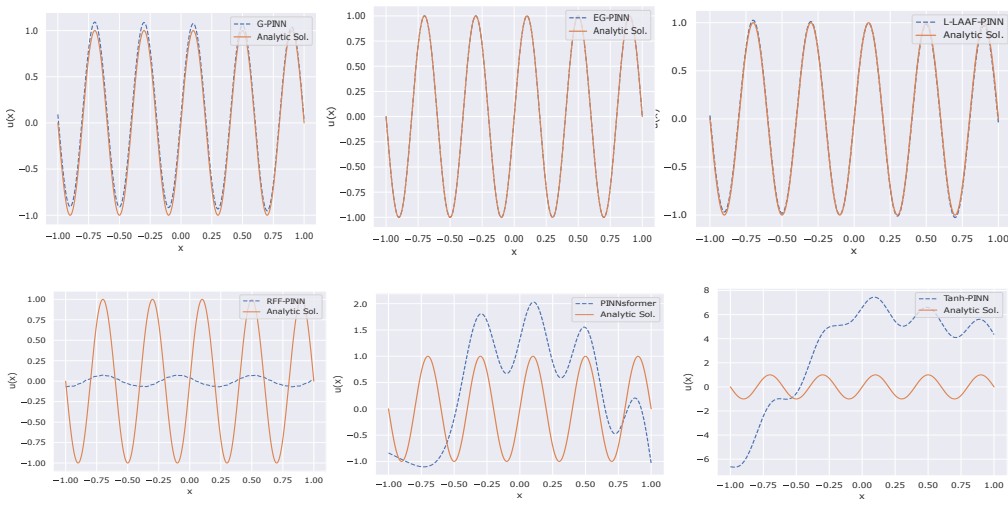

Figure 15: Reconstruction of all networks after training for only 5000 epochs to solve Poisson's equation. EG-PINN has found an extremely good fit in such a small number epochs. G-PINN and L-LAAF have found a decent fit and RFF-PINN, PINNsformer and Tanh-PINN seem to have found only a low frequency solution.

