# OpenReview forum: "Architectural Insights for efficient Physics-Informed Neural Network optimization"
_ICLR.cc/2024/Conference — ICLR 2024 Conference Withdrawn Submission_

### Official Review · Reviewer_Az9b · 2023-10-15

**Soundness:** 3 good
**Presentation:** 2 fair
**Contribution:** 2 fair
**Rating:** 5
**Confidence:** 4

**Summary:**

The manuscript studies optimization issues in the context of Physically Informed Neural Networks, specifically ones with Gaussian activation functions. The paper contains several related results.

1. Capitalizing on previous results showing that the lowest NTK eigenvalue correlates with convergence--- the authors derive a lower bound on the lowest NTK eigenvalue for the specific case of Gaussian activations. They argue that it increases favourably with width thereby allowing convergence for smaller networks.

2. They then turn to study the conditioning number (ratio of highest to lowest eigenvalue) of a relevant part of the Loss Hessian and suggest a method of improving this number by evening out (equilibrating)  the rows of weight matrices.

3. They perform several experiments supporting their statements that Gaussian PINNs and more so Equilibrated Gaussian PINNs are optimal for performance as well as better conditioned.

**Strengths:**

The paper applies recent results related to NTK to an important relatively fresh domain, that of PINNs.

The theoretical effort is grounded and closely tied with experiments.

It provides a novel PINN normalization scheme which may lead to improved performance.

**Weaknesses:**

The work is somewhat of a mixed bag of results. We have what is hopefully an exact result on the scaling of the lowest eigenvalue, a heuristic discussion of conditioning numbers of the Hessian, and experiments which partially support apparently known claims (that G-PINNs are better) and some which support the results on conditioning numbers and EG-PINNs. Due to this lack of cohesiveness, I feel that, to some extent, each of these three results should stand on its own.

For example, the conditioning number is huge for all three architectures in Fig. 4. so can one really trust it to be an indicator of performance? If so should I expect EG-PINN performance to be 10^5 better than the rest (it isn't...)? If I try and rely on section 4. I encounter many heuristics and if I want to base this on the first result, then that first section discusses the lowest eigenvalue rather than the conditioning number and furthermore does that for a different matrix.

More specifically I think potential weaknesses are the following
1. More numerics is required to support the claim that EG-PINN is a novel competitive architecture. For instance, if this was a claim on an image classifier, showing superior results on CIFAR-10, Fashion-MNIST, and ImageNet would have been the bare minimum for making such a claim.

2. As NTK spectral is typically highly multiscale (and very loosely related to the Hessian), I'm not surprised by the huge conditioning numbers the authors find. However, I find it hard to see how such huge numbers can affect experiments. More probable in my mind, is that only the high Hessian modes are relevant and perhaps those are correlated with the lowest ones. To make this a bit more concrete say that we study standard NTK dynamics and that this conditioning number is the NTK conditioning number. This would mean the lowest eigenmodes of the target would be learned 10^30 times slower than the highest ones. Moreover, the learning rate is typically adjusted w.r.t. to the higher modes as they produce the majority of the gradients. Hence a realistic learning rate is one where the highest modes are learned in O(1) epochs hence the lowest ones in O(10^30) epochs... To summarize a better link is required between the heuristic claims of Sec. 4. and the experimental results.

3. Generally, for properly normalized neural networks, the NTK eigenvalues scale as N. Given that the authors take n_k > N, I can't see how the statement that \lambda_{min}(K) is asymptotically dominating n_k^2 can be true. It appears that the authors are following the notations of previous works but still, this needs to be clarified.

4. Evidence for the scaling of point 3. shown in Fig. 1. is quite weak. To show power law behaviour, I'd expect to see a log-log plot with at least two orders of magnitude. Similarly

5. The theory of the current work is not sharply dependent on the PINN setting. Why are the authors allowed to analyze just the boundary aspects of the PINN NTK? From my experience, and from the authors' data, the boundary behaviour is typically easier to capture than the bulk behaviour. Would they argue that Gaussian activation is also good for image classification?

**Questions:**

Please see previous weaknesses section.

---

> ### Author Response · Authors · 2023-11-20
> **Response to reviewer Az9b**
>
> We thank the reviewer for their review and their questions. We apologize for taking time to post our rebuttal but the reviewer had asked questions that required us to undertake further experiments so as to answer the reviewers questions. We hope the reviewer will be understanding about this.
>
> **The work is somewhat of a mixed bag of results. We have what is hopefully an exact result on the scaling of the lowest eigenvalue, a heuristic discussion of conditioning numbers of the Hessian....**
>
> The discussion on condition numbers in the main paper is heuristic as ICLR only offer 9 pages so we cannot go into depth with mathematical results. However, in appendix section A.3 (of the original submission) we clearly showed that many of these heuristic are right by proving several mathematical statements about conditioning.
>
> **Due to this lack of cohesiveness, I feel that, to some extent, each of these three results should stand on its own...**
>
> First of all, rehashing a paper as 3 separate papers because the original paper had 3 results is worrying to say the least. The goal of machine learning research is not paper churning but rather to expand the current knowledge within the community. It is clear you have not understood the merit in putting these 3 results together so we think it is important to take the time to explain it to you: Given a feed forward
> neural network what are the main components of the network? There are two main components, 1. the linear structure which consists of the weights and biases, and 2. the activation which is a non-linear function. Thus in order to design a  an effective feedfoward neural network architecture, one has to consider two main architectural perspectives. Namely, the linear structure and the non-linear structure. We show that in order to design network with effective non-linear structures, the NTK theory gives a good insight as to suggesting what sort of activation to choose. We show that in this regard a Gaussian activation is optimal from the viewpoint of the NTK. We then move on to considering the linear structure and we show that numerical linear algebra techniques offer a good insight as to how one should normalize the weights. We show the best technique in this regard is equilibration. Theoretically we show why this is the case in appendix sec. A.3, and then we back this up with experiments. This is the point of the paper and it clearly shows how each section is connected.

---

> ### Author Response · Authors · 2023-11-20
> **Response to reviewer Az9b (part 2)**
>
> **For example, the conditioning number is huge for all three architectures in Fig. 4. so can one really trust it to be an indicator of performance? If so should I expect EG-PINN performance to be 10^5 better than the rest (it isn't...)? If I try and rely on section 4. I encounter many heuristics and if I want to base this on the first result, then that first section discusses the lowest eigenvalue rather than the conditioning number and furthermore does that for a different matrix..**
>
> Again you have completely missed the point of the paper. Designing a feedforward network comes down to two main concepts. The non-linear structure and the linear structure. We show that to design an effective non-linear structure one should go through NTK. For this the lowest eigenvalue of the NTK matrix is important and so we seek a non-linearity that gives the best minimum eigenvalue scaling for the NTK matrix. In order to define effective linear structures for optimizing a network efficiently we show that the best way forward is to look at the weight conditioning and how it is related to the Hessian. The Hessian is now a different matrix, we are looking at the Hessian of the loss function (the NTK only looks at the model structure) and we show how equlibration can help with condititioning of the Hessian. What this means is it helps with making the Hessian smoother. If you are not happy with heuristics you need to read appendix section A.3 where clear mathematical statements are proved. We clearly reference this in the paper asking the interested reader to consult that section.
>
> **More numerics is required to support the claim that EG-PINN is a novel competitive architecture. For instance, if this was a claim on an image classifier, showing superior results on CIFAR-10, Fashion-MNIST, and ImageNet would have been the bare minimum for making such a claim.**
>
> This is a very unfair comment. There are several papers on image classification that only do experiments on MNIST (not even Fashion-MNIST) and are published in top tier conferences. As an example please see paper [1] which was an oral in CVPR 2022. It only does experiments on MNIST. A simple Google will show you many many more papers on image classification that are published in top tier conferences that only do one or two data sets. Furthermore, this is not just true for image classification but also true in regeression papers. The paper [2]
> only studies one dataset for neural radiance fields and was an oral in ECCV 2022. So the standpoint that one needs to do all of CIFAR-10, Fashion-MNIST, and ImageNet is completely false claim.
> The point is not simply about running all sorts of experiments the point is that this paper develops theoretical results in full and gives their proof in full in the appendix. Did you read the appendix? If you are not happy with the heuristics in the main paper you should see the appendix where detailed proofs and mathematical statements are given. Did you read those statements and proofs? They answer most of your questions.
>
> Furthermore, in total we have given four different experiments. We have given three in the main paper and one in the appendix. Did you see appendix A.8 in the original submission of the paper where another experiment is given? Furthermore, all our experiments are of very different nature and of different dimensions. We do the 1d, 2d and 3d case. However, let us take a look at whether the reviewer is right in what they claim is a lack of numerics by comparing with some of the most cited papers in the PINN literature. Paper [3] only experiments with 2 pdes (211 citations), paper [4] only experiment with 2 pdes (481 citations), paper [5] only experiments with 3 pdes (210 citations). In our conference paper we are doing a total 4 different pdes (please see appendix A.8 in the original submission for the fourth pde).
>
> [1] L. MacDonald et al. Enabling equivariance for arbitrary lie groups; CVPR 2022.
> [2], S. Ramasinghe et al. Beyond periodicity: Towards a unifying framework for activations in coordinate-mlps; ECCV 2022
>
> [3] S. Wang et al. On the eigenvector bias of Fourier feature networks: From regression to solving multi-scale PDEs with physics-informed neural networks; Computer Methods in Applied Mechanics and Engineering, vol. 384.
>
> [4] S. Wang et al. When and why PINNs fail to train: A neural tangent kernel perspective; 	arXiv:2007.14527
>
> [5] L. McClenny et al. Self-adaptive physics-informed neural networks using a soft attention mechanism; arXiv:2009.04544

---

> ### Author Response · Authors · 2023-11-20
> **Response to reviewer Az9b (part 3)**
>
> **As NTK spectral is typically highly multiscale (and very loosely related to the Hessian), I'm not surprised by the huge conditioning numbers the authors find. However, I find it hard to see how such huge numbers can affect experiments. More probable in my mind, is that only the high Hessian modes are relevant and perhaps those are correlated with the lowest ones....**
>
> Would you please provide us with a reference to back up your claims? As a reviewer it is important for you to initiate a discussion between yourself and the authors. You are clearly not doing this in any way. You are simply writing statement you believe to be true without any references or citations to results. You say ``More probable in my mind, is that....". With all due respect what is probable in your mind is really not helpful in terms of a discussion. You need to tell us why this is probable and give us a citation to show us that
> what you say is indeed correct. Unfortunately what you say is completely wrong. The higher Hessian modes are not the ones that are relevant. The highest Hessian mode is relevant when working out a bound on the learning rate needed for an iterative descent algorithm, please see [1] where this is clearly explained. The bound on the learning rate is given so that a convergence theorem can be proved and this is only useful in the convex setting. However, once a learning rate bound is found, the question that arises is how fast can convergence to a minimum occur? For this a complexity theorem can be proven which shows that for the convex setting the complexity rate is linear and for non-convex in general it is sub-linear. Once this is established one can still pose the question, are there ways to speed up the convergence within the complexity class? This where the condition number comes in. The point is that you want to understand the rate of progress at each iteration of the algorithm in each eigenvector direction of the Hessian. This is bounded above by the condition number and it is this bound that tells one that if the condition number is big then the rate of progress in that given direction is small, if the condition number is small then the rate of progress if bigger. If you do not understand this please see these lecture notes, p. 25, where it is clearly explained: https://www.cs.toronto.edu/~rgrosse/courses/csc421_2019/slides/lec07.pdf
>
> **Generally, for properly normalized neural networks, the NTK eigenvalues scale as N. Given that the authors take n_k > N, I can't see how the statement that \lambda_{min}(K) is asymptotically dominating n_k^2 can be true. It appears that the authors are following the notations of previous works but still, this needs to be clarified.**
>
> The reviewer needs to provide a reference for this. Again you are really not initiating any discussion but simply stating facts that you percieve contradict our work when they don't. What you say is only true for ReLU networks. In the case the network employs a ReLU activation then it is true that the NTK eigenvalues scale linearly with N. A reference to this was already given in the original submission of the paper. The paragraph following thm. 3.1 we clearly explain this with citations.
>
> **Evidence for the scaling of point 3. shown in Fig. 1. is quite weak. To show power law behaviour, I'd expect to see a log-log plot with at least two orders of magnitude.**
>
> Thank you. However, we are sticking with what people do in the NTK community as they are our main audience. Furthermore, people in the intersection of the PINN and NTK community generally plots the eigenvalues in this way. We have uploaded a revised version of this paper where we have added the scaling of the tanh network in Figure 1. You can clearly see that the Gaussian network scales much higher.

---

> ### Author Response · Authors · 2023-11-20
> **Response to reviewer Az9b (part 4)**
>
> **The theory of the current work is not sharply dependent on the PINN setting. Why are the authors allowed to analyze just the boundary aspects of the PINN NTK?....**
>
> This is a very good question and we understand the reviewers concern.
> We did clearly state it in the "limitations" section where we concretely stated that one of the limitations of the work is that we were not able to obtain a scaling law for the pde residual term. What makes this difficult is that one needs to understand the theory of the NTK of derivatives of the network, which is much more harder than simply understanding the NTK of the network alone. Please have a look at the analysis in the appendix where it is shown even in the case of the NTK of the model how difficult the probability anylsis is. Extending this to the case of derivatives of the model is a very good question but is out of the scope of this work. So far, as we are aware no one in the machine learning literature has been able to obtain scaling laws for the NTK for derivatives of the network.
> However, there is still merit in our approach. As can be seen from experiment 1 applying a Gaussian not only influenced the boundary loss error but it also influenced the pde loss term. This suggest that the NTK of the pde term of the Gaussian network is also being affected in a positive way. This is only empirical though and as of yet we have no way to prove this theoretically. That being said research is incremental so it is important to understand that one cannot achieve everything in one go. We believe understanding how the NTK scaling is affected by the pde term is an important future work that the community should look into.

---

> ### Comment · Reviewer_Az9b · 2023-11-21
> **Reply to the authors' reply**
>
> I read the authors' reply thoroughly, despite the difficulty of doing so. Unfortunately, I haven't found much more than a lecturing and occasionally aggressive tune. This was also prevalent in their reply to the rest of the referees. I encourage the authors to adopt a more scientific mode of discussion and even more so, take any alleged failing in a reader's understanding of their work (not that I think there was one) as a pointer to where they may improve their presentation. Other than that I read their points but see no reason to raise my score.
>
> p.s.
>
> Regarding NTK having a multi-scaled spectrum see for instance Fig. 3 here https://arxiv.org/pdf/2007.14527.pdf

---

> > ### Author Response · Authors · 2023-11-21
> >
> > Thank you. We will definitely take your comments and improve the presentation of the paper.
> >
> > You miss understood our question. We were not asking for a citation for NTK having multihead spectrum. We were asking for a citation for your comment "More probable in my mind, is that only the high Hessian modes are relevant". Could you provided a reference for this? As this will definitely help us improve our presentation and allow us to cite necessary results that show that only the high modes of the Hessian are relevant.

---

### Official Review · Reviewer_5UPq · 2023-10-19

**Soundness:** 2 fair
**Presentation:** 2 fair
**Contribution:** 3 good
**Rating:** 3
**Confidence:** 3

**Summary:**

The paper shows that the smallest eigenvalue of the NTK of a PINN's data loss can be bounded from below if the activation functions are Gaussian, and that the condition number can be reduced by preconditioning of the initial weight matrices (row equilibration). These architectural changes -- equilibration and Gaussian activations -- are then shown to improve training success and test performance.

**Strengths:**

The paper treats an interesting and timely topic, as the training of PINNs is still not fully understood. From this perspective, the results in the paper are important contributions to a better understanding of PINNs. At a first glance, the experiments seem to support the theory presented in the paper.

**Weaknesses:**

The paper has, in my opinion, not matured sufficiently to merit publication at an A* venue, due to the following weaknesses:
* The experiments seem to rely on a setting different from the theory. While the theoretical results in Th. 3.1 require that the layer width $n_k$ is larger than $N$ (the number of collocation points, I assume), the experiments use comparably small layer widths (128 neurons). On the one hand, the theory is thus applicable to only very narrow settings rarely used in practice. On the other hand, it is then less clear how the theory and experimental evidence are connected, since the theorem does not hold for the experimental setting. From that perspective, I would be interested in seeing results also for PDEs exhibiting periodic behavior, to investigate whether the Gaussian activation still outperforms the sine activation.
* For the experiments, an ablation study is missing. If I understood correctly, 5.1 and 5.2 do not employ equilibration but only Gaussian activations, while 5.3 uses both, but does not show results for equilibration only (i.e., without Gaussian activations). I would appreciate seeing results (e.g., in 5.1) for the G-PINN, EG-PINN, and a PINN that uses tanh activations but preconditioned weight matrices.
* The results are not fully convincing. E.g., in Fig. 1 the quartic increase is not evident and would require much larger $N$. Also, the curve for tanh seems to be constant, while theory suggests quadratic behavior. Can you explain this discrepancy?
* The notation is not fully explained or clear (see below).
* The experimental setup is not fully clear: E.g., in Sec. 5.2, it is not clear how the initial and boundary conditions were chosen and if simulation data was available for training or not.
* Some statements in the paper are redundant, e.g., the last sentence on page 2, the first paragraph on page 4. The last paragraph of Sec. 4.1 is not fully clear. The font size in Fig. 2 is too small.

**Questions:**

See above for the most critical questions. Additional questions:
* Does $N$ in Th. 3.1 refer to $N_b$ or to $N_r$?
* How is the $k$ in Th. 3.1 related to the number $L$ of layers?
* What is the $\beta_k$ in Th. 3.1?
* In Prop. 4.2, how does equilibration affect the first term on the right-hand side of the inequality?
* Is there a connection between Gaussian activations and equilibration, or are these two independent ingredients?
* In Table 3, does the IC error and PDE error refer to test or training data?

---

> ### Author Response · Authors · 2023-11-20
> **Response to reviewer 5UPq**
>
> We thank the reviewer for their review and their questions. We apologize for taking time to post our rebuttal but the reviewer had asked questions that required us to undertake further experiments so as to answer the reviewers questions. We hope the reviewer will be understanding about this.
>
> **The experiments seem to rely on a setting different from the theory. While the theoretical results in Th. 3.1 require that the layer width...**
>
> This is always the case with NTK theory. In general NTK theory is only applicable in the infinite width setting, which is impossible to use practical. However, to suggest therefore that NTK theory is useless is a complete misunderstanding of what it means to do mathematical theory for an area of science. The purpose of theory is to give insight to the practitioners of the field so as to guide them to design better networks and ask research questions that enlighten the community. The NTK theory since it came out has done just this. The original NTK paper [1] has roughly 2610 citations, yet its result are only applicable in the infinite width setting. Are you then saying that therefore it is a useless paper for practitioners in machine learning? Of course not. The paper has given the ML community significant insight and although it holds only in the infinite width setting it has still yielded great results even in practical ML. The purpose of this paper is likewise to give insight based on theory. This can never be perfect and we don't claim it is yet when we test our insight we see practically that the Gaussian PINN is doing far better than the others. Furthermore, your understanding of overparameterized settings is incorrect. A large width network can be seen as the limit of a small width network. This means that the small width networks gradient descent dynamics can be equated to the large widths networks gradient descent dynamics plus an error term. The reason why NTK has become so important is that people are finding that the error term is significantly small that the insight gained from the large width dynamics seems to give a good approximation to the dynamics to the small width dynamics. This is the whole point of why NTK theory is so useful. If you do not understand this you should really read the original NTK paper [1].
> Furthermore, this insight has been used in several PINN papers such as [2] (211) citations, [3] (481 citations), just to reference a few.
>
>
> [1] Neural Tangent Kernel: Convergence and Generalization in Neural Networks; 	arXiv:1806.07572 [cs.LG]
>
> [2] On the eigenvector bias of Fourier feature networks: From regression to solving multi-scale PDEs with physics-informed neural networks; Computer Methods in Applied Mechanics and Engineering, vol. 384.
>
> [3] When and why PINNs fail to train: A neural tangent kernel perspective; 	arXiv:2007.14527 [cs.LG]
>
> **would be interested in seeing results also for PDEs exhibiting periodic behavior, to investigate whether the Gaussian activation still outperforms the sine activation.**
>
> We did a pde exhibiting periodic behaviour in the appendix A.8 in the original solution. We thank the reviewer for suggesting we include a result with a sine activation for this. Please see the following results.
> As can be seen the Gauss network does marginally better. Furthermore, it is well known that sine activated neural networks can perform very unstably for various ML problem, see [1], and we noticed over different pdes a sine activated pde also performed unstably.
> Given that there are a lot of pdes that are not periodic, our results show that Gaussian is a much better activation to use within the broad context of modelling pdes.
>
> |             | Train error | Rel. test error | Bndry. error | Pde error | Time (s) |
> |-------------|-------------|------------------|--------------|-----------|----------|
> | G-PINN      | $1.31e-2$   | $1.15e-3$        | $4.72e-3$    | $8.52e-3$ | 20.01    |
> | EG-PINN     | $6.0e-4$    | $4.97e-7$        | $8.51e-7$    | $6.14e-7$ | 98.95    |
> | L-LAAF-PINN | $7.98e-3$   | $9.06e-4$        | $1.38e-3$    | $6.61e-3$ | 20.22    |
> | PINNsformer | $1.46e2$    | $2.42e0$         | $6.18e-4$    | $1.4e2$   | 267.13   |
> | RFF-PINN    | $3.05e-4$   | $1.01e0$         | $1.88e-3$    | $3.15e4$  | 39.25    |
> | Tanh-PINN   | $9.33e2$    | $4.87e1$         | $3.11e1$     | $9.04e2$  | 13.78    |
> | Sine-PINN   | $1.36e-2$   | $1.17e-3$        | $4.71e-3$    | $9.34e-3$ | 17.54    |
>
> It is not surprising that the Gaussian does better as it had already been shown in [1] that sine activated networks can be unstable when training with descent algorithms but Gaussians are much more stable.
>
> [1] S. Ramasinghe et al. Beyond periodicity: Towards a unifying framework for activations in coordinate-mlps; ECCV 2022

---

> > ### Comment · Reviewer_5UPq · 2023-11-21
> > **Reply**
> >
> > Thank for your response.
> >
> > *ad NTK*: I am well aware that the NTK regime is a limiting regime that is not achievable in practice. I am, indeed, more worried about the fact that the theorem is based on a neurons-to-data ratio much larger than 1, while the experiments are based on a ratio much smaller. In this particular setting, I am not convinced that the error term that you speak of is negligibe. Maybe you could try to use (much) less than 128 collocation points to fall within the scope of the theorem? (I realize, though, that this may make PINN training infeasible.)
> >
> > *ad sine activation/ablation study/independent ingredients*: It is very interesting to see that the Gaussian activation function outperforms the sine activation function even for periodic solutions, and that equilibration does not lead to superior performance for tanh activation functions, thanks for the experiments. I was, by the way, not asking for "ablation after ablation", but just for particular one. Since you confirmed my suspicion that Gaussian activation function and equilibration are independent ingredients (the paper was not sufficiently clear on this, in my opinion), the ablation study I asked for (equilibration for non-Gaussian activation functions) is very reasonable.
> >
> > *train/test error*: It is definitely not the case that in the PINN community always the training error is reported. Much more meaningful in my opinion (and also reported by some PINN papers) are measures like some relative (absolute/squared) error evaluated on collocation points sampled independently from the training collocation points. After all, we are interested in how the PINN generalizes to points not seen during training.
> >
> > *Experimental Setup": Most studies on training problems of PINNs focus on the forward problem, i.e., a problem where the initial and boundary conditions are given and where the task is to learn the solution of the PDE in the interior of the computational domain. In these settings, not data (simulation results) are available. Situations like this have been studied by various authors (see, e.g., the Burgers' equation in the original PINN paper), and simulation data is then only used to evaluate the error. The setup you studied in your work is substantially simpler, as the PINN can access training data in the interior of the computational domain. I am not saying that the training problem becomes trivial (it is still a multi-objective optimization problem with a potentially complicated landscape) -- but note that this problem is practically even less relevant (it is neither an inverse problem nor a forward problem).
> >
> > *Appendix:* Regarding the appendices: As a reader/reviewer I am not obliged to search the questions to my answers in the appendix. Rather, I guess one can expect the authors to point to all appendices that may be relevant for the reader. A simple sentence such as "The experimental setup (data generation, architecture, etc.) is explained in Appendix A.6." would suffice.
> >
> > At the moment I don't see any reason to change my score, but I am happy to continue the discussion.

---

> > > ### Author Response · Authors · 2023-11-21
> > >
> > > Thanks for your response.
> > >
> > > **ad NTK: I am well aware...**
> > >
> > > 128 collocation points is simply too little for PINN training. What would be more helpful is if you gave a citation to a few PINN papers that uses NTK that works in the regime you are wanting. Almost all PINN papers use NTK theory, which is for highly overparamterized regimes, and then train on much smaller architectures i.e. neurons to data ratio being smaller than 1. That way we can look at those papers and try to better our result for a future submission. Could you provide us with one or two citations?
> > >
> > > **ad sine activation/ablation study/independent ingredients: It is very interesting**
> > >
> > > Not sure how you are interpreting the results. But equilibration does help the tanh network. If you compare the tanh network and the tanh network with equilibration (ETanh) you can see that ETanh does better. In fact if you look at all the networks without equilibration and then you compare each one with euilibration, you see from the results that the equilibrated ones do better.
> > >
> > > **train/test error:**
> > >
> > > With all due respect we have given both errors. We have given train errors for collocation points and boundary/initial points. And then we have given relative test errors on on a bigger sample of points where we sampled both collocation and bndry/initial points. These test errors are relative squared errors.
> > >
> > > ** *Experimental Setup**
> > >
> > > This is exactly what we are doing. Which experiment are you talking about? In the training pass we use the actual values only for the initial and bndry points. For the pde residual we sample collocation points. Then when we test, we sample the actual values and use this to compare, this forming the relative test error. We are not using any interior test label for training. Please show us which experiment you are talking about and where we used actual labels for training interior points.

---

> > > > ### Comment · Reviewer_5UPq · 2023-11-21
> > > >
> > > > *Not sure how you are interpreting the results. But equilibration does help the tanh network.*: Yes, it does, and that was worth showing, right? (So it's not just to make the reviewer happy.) My comment that ETanh-PINN is not superior was only pointing to EG-PINN being better, thus somehow justifying your statement " As our theory predicts equilibrating does help all PINNs but EG-PINN still performs the best."
> > > >
> > > > *In the training pass we use the actual values only for the initial and bndry points. For the pde residual we sample collocation points.*: Then why do you write in the response: "Also, what do you mean initial and boundary conditions? This is a Navier-Stokes equations trained only on full pde data." This is what I meant: The experimental setup is not clear.
> > > >
> > > > *These test errors are relative squared errors.*: Indeed, I have overlooked that and misunderstood your response. I'm sorry for that slip.

---

> > > > > ### Author Response · Authors · 2023-11-21
> > > > >
> > > > > **Yes, it does, and that was worth showing, right?...**
> > > > >
> > > > > This is very reasonable to say. The only reason we did not include it in the original paper was due to space constraints. We struggled to fit three distinct pdes in the main paper and this caused us to focus on the best architectures. However, we are very happy to put these tables in the appendix. It definitely does show that equilibration and helps and is useful info for the reader. Thank you for pointing that out.
> > > > >
> > > > > **Then why do you write in the response:...**
> > > > >
> > > > > Ok I see the problem. There is confusion because the way people train Navier Stokes is different. For Navier stokes there is no actual boundary term or initial term. What we are training on is the velocity term, this is the supervised part and this replaces the boundary/initial condition term. The second part of the training of the pde loss term which is unsupervised. For the velocity term we randomly sample points in the whole domain and we have the actual labels there as well from simulated data. So we can do supervised training for the velocity term over those sampled points. For the pde term, we sample random collocation points in the interior and we use those for the pde loss in an unsupervised fashion.
> > > > > Now we come to table 2 in the paper. The first column which is the L^2 train error is the total loss consisting of the supervised velocity error over the velocity training points + the unsupervised pde error over the collocation points. We then reported test errors in the next columns. The relative test pressure is computed and tested against simulated data. The relative test velocity can also be computed because we had simulated data that gave the velocity labels as well, though when testing we use a different data set to the train data set for velocity. Does this make sense? The original paper does exactly this and we are using the simulation data from that paper, so the date we are using is data used by others.

---

> > > > > > ### Comment · Reviewer_5UPq · 2023-11-21
> > > > > >
> > > > > > _For the velocity term we randomly sample points in the whole domain and we have the actual labels there as well from simulated data._ -- Exactly, and this is a much simpler problem as the forward problem. (The forward problem was studied in the original PINN paper, while a problem similar to yours was studied in the "Hidden Fluid Mechanics" paper by the same authors.) Note that also in fluid dynamics you can study problems with initial and boundary conditions.
> > > > > >
> > > > > > That said, I think the confusion is now evident, and it should be obvious that a more careful description (major revision) of the experimental setup is needed.

---

> > > > > > > ### Author Response · Authors · 2023-11-21
> > > > > > >
> > > > > > > **Exactly, and this is a much simpler problem.../The forward problem was studied in the original PINN paper...**
> > > > > > >
> > > > > > > Sigh. The "Hidden Fluid Mechanics" uses both a physics uninformed network together with a physics informed network. Equation (1) of that paper shows what they are doing and how their training is proceeding. We are not doing that. We are doing exactly what they are doing in the original PINN paper with the difference being that we are not using noisy samples and we are replacing boundary/initial data with velocity data, this is done so that the pde equation is not ill posed. We are then extracting the pressure at the end to show that even though no training was done on pressure you can still approximate the pressure solution with a forward method. This is what the forward approach does. There was not training on the pressure solution. I never said in fluid dynamics you cannot study problems with initial and boundary conditions what I said was that for Navier-Stokes you don't need to take that approach. What I meant by this is that you can instead replace that data with velocity data and still get a well-posed pde problem whose solution you can then approximate with a neural network, which is a forward problem.
> > > > > > >
> > > > > > > In any case, we have to agree to disagree otherwise we will be here forever. Thanks for your time though. Really appreciate it. We took for granted that people would understand the experimental setup. We will explain this much more thoroughly in future revisions. Thank you.

---

> ### Author Response · Authors · 2023-11-20
> **Response to reviewer 5UPq (part 2)**
>
> **For the experiments, an ablation study is missing. If I understood correctly, 5.1 and 5.2 do not employ equilibration but only Gaussian activations, while 5.3 uses both, but does not show results for equilibration only (i.e., without Gaussian activations). I would appreciate seeing results (e.g., in 5.1) for the G-PINN, EG-PINN, and a PINN that uses tanh activations but preconditioned weight matrices.**
>
> The reason we did not include a result where we equilibrate a Tanh PINN is because a normal G-PINN does better. So we went with the best. You also have to understand ICLR only allows 9 pages. We developed theory in full for the use of Gaussian via NTK and also developed theory as to why equilibrating is a good idea. On top of this we simply don't have the space to run ablation after ablation trying to explain every single result. That's not the purpose of a conference and is more suited towards a journal. Even in the case of a journal several PINN papers only consider two pdes e.g. [2], [3] yet in total we did 4. 3 in the main paper and 1 in the appendix (see A.8). However, so as to make the reviewer happy please see the results of equilibrating Tanh PINN, Gaussian PINN and sine PINN for Bugers' equation. As our theory predicts equilibrating does help all PINNs but EG-PINN still performs the best. The notation for the table below, is ETanh-PINN denotes an equilibrated Tanh PINN, EG-PINN denotes equilibrated Gaussian PINN and ESine-PINN denotes equilibrated sine PINN.
>
> |            | Train error | Rel. test error | Time (s) |
> |------------|-------------|------------------|----------|
> | G-PINN     | $1.61e-3$   | $1.25e-2$        | 92.78    |
> | Sine-PINN  | $5.36e-2$   | $2.31e-1$        | 60.79    |
> | Tanh-PINN  | $9.81e-3$   | $4.12e-2$        | 53.48    |
> | EG-PINN    | $1.29e-4$   | $1.11e-3$        | 211.23   |
> | ESine-PINN | $8.45e-3$   | $9.89e-2$        | 154.45   |
> | ETanh-PINN | $1.73e-3$   | $7.36e-2$        | 113.24   |
>
> **The results are not fully convincing. E.g., in Fig. 1 the quartic increase is not evident......
> Also, the curve for tanh seems to be constant, while theory suggests quadratic behavior. Can you explain this discrepancy?**
>
> We plotted the minimum eigenvalue as the width of the network scales by $8N$, where $N = 400$ is the fixed number of data samples. Then plotted this against the curve $\frac{1}{3\times 10^{10}}(8N)^4$.
>
> The curve for tanh is not constant. The reason it looks constant in the figure is because the minimum eigenvalue of the Gaussian network scales much more larger than the tanh network one. We have added a zoom in of the tanh network scaling in figure 1 in the revised upload of the paper. Please see the right side of the figure. It is clear the tanh network scaling is not constant.
>
> **The experimental setup is not fully clear: E.g., in Sec. 5.2, it is not clear how the initial and boundary conditions were chosen and if simulation data was available for training or not.**
>
> This has clearly been explained in the appendix in section A.6 of the original submission. We clearly explained where we got the data from. We also explained that it is synthetic data and even gave the website we got the data from. We also explained how the points were sampled, the optimizer used batch size, number of epochs etc.
>
> Also, what do you mean initial and boundary conditions? This is a Navier-Stokes equations trained only on full pde data. That is why in the experiment there are no results on boundary or initial condition train errors. This is a standard example in the fluid dynamics community. We also gave the explicit reference [1] of the paper we are testing against. The ICLR conference is a 9 page conference, therefore it is unreasonable to expect us to explain every single detail in the main paper. If you have questions regarding extra details you should read the appendix and check if it answers what you want. When it doesn't you should then let us know and we can give you any extra information you need. Furthermore, we cannot explain the basis of fluid dynamics modelling in a conference paper. If you have never done any work on Navier-Stokes equations then it is highly recommended for you to consult reference [1]. However, it is unreasonable to expect authors of a conference paper to redo work that is already done in a paper. How to model the Navier stoked equations and how to set them up as a PINN is all detailed in [1].
>
> [1] M. Raissi et al; Physics-informed neural networks: A deep learning framework for solving forward and inverse problems involving nonlinear partial differential equations. Journal of Computational physics,378:686–707,2019.

---

> ### Author Response · Authors · 2023-11-20
> **Response to reviewer 5UPq (part 3)**
>
> **Some statements in the paper are redundant, e.g., the last sentence on page 2**
>
> We have taken out these redundancies for you. Please see updated revision.
>
> It would be much more helpful if you told us what exactly is not clear about the last paragraph in section 4.1. Could you please tell us what it is you don't understand?
>
> **Does $N$ in Th. 3.1 refer to $N_b$ or to $N_r$?**
>
> Theorem 3.1 holds for any feed forward neural network trained by MSE loss. Therefore, the theorem is independent of whether the network is a PINN or not. We then apply it to the setting of PINNs. In the setting of PINNs, the $N$ from thm. 3.1 is $N_b$.
>
> **How is the k in Th. 3.1 related to the number of layers?**
>
> This is a typo and we thank the reviewer for pointing it out. The number of layer is fixed as $L$ and the k in the statement of the theorem is supposed to reference those layer before the final layer, so $1 \leq k \leq L-1$. We have corrected the notation of the theorem. Please see revised upload.
>
> **What is  the $\beta_k$ in Th. 3.1?**
>
> This is the variance of the Gaussian distribution that is being used to initialize the weights of the kth layer. As we mentioned before the statement of thm. 3.1, \textit{``We will now give a synopsis of the
> theorem but kindly ask the reader to consult appendix A.1 for detailed notations, assumptions and
> proofs"}. The notation is clearly explained in the appendix in section A.1. The reason for putting the notation there is because we only have 9 pages and going through all the notation and assumptions will take a lot of space. We kindly ask the reviewer to see sec. A.1 for more details.
>
> **In Prop. 4.2, how does equilibration affect the first term on the right-hand side of the inequality?**
>
> The first term on the right hand side of the inequality is a diagonal matrix. Therefore, equilibrating it does not affect the condition number. This is a simple result from Linear algebra. A diagonal matrix has its eigenvalues already on the diagonal. If you equilibrate you are simply making the diagonal terms have norm 1. If you then take the condition number the norms cancel out as you are doing largest eigenvalue divided by smallest. Here we are assuming non of the diagonal entries are zero. Note that for a Gaussian activation this is indeed the case with high probability as the Gaussian has zero set a set of measure zero.
>
> **Is there a connection between Gaussian activations and equilibration, or are these two independent ingredients?**
>
> These are independent. You can equilibrated any network you like. Though if you are asking this question we feel you have not understood the paper. So let us take the time to explain the point of the paper to you. Given a feed forward
> neural network what are the main components of the network? There are two main components, 1. the linear structure which consists of the weights and biases, and 2. the activation which is a non-linear function. Thus in order to design a  an effective feedfoward neural network architecture, one has to consider two main architectural perspectives. Namely, the linear structure and the non-linear structure. We show that in order to design network with effective non-linear structures, the NTK theory gives a good insight as to suggesting what sort of activation to choose. We show that in this regard a Gaussian activation is optimal from the viewpoint of the NTK. We then move on to considering the linear structure and we show that numerical linear algebra techniques offer a good insight as to how one should normalize the weights. We show the best technique in this regard is equilibration. Theoretically we show why this is the case in appendix sec. A.3, and then we back this up with experiments.
>
> **In Table 3, does the IC error and PDE error refer to test or training data?**
>
> This is always train error as is standard in the PINN community. Testing is done on a whole batch that includes IC and PDE points together.
>
> **The paper has, in my opinion, not matured sufficiently to merit publication at an A$*$ venue, due to the following weaknesses**
>
> This comment is very disappointing from the reviewer. Many of the weaknesses you have brought up are really weaknesses in your own understanding and yet you are using that to put down this paper.

---

### Official Review · Reviewer_vRbU · 2023-10-27

**Soundness:** 2 fair
**Presentation:** 2 fair
**Contribution:** 2 fair
**Rating:** 3
**Confidence:** 4

**Summary:**

The paper improves PINN in (1) Gaussian activations and (2) an architecture that conditions neural weights.

**Strengths:**

The paper is well-motivated and did rigorous theoretical analysis for both the Gaussian activations and the architecture that conditions neural weights..

**Weaknesses:**

Activation functions have been well-studied in the PINN literature: https://arxiv.org/abs/2209.02681.
Nowadays, proposing new activations does not seem to be novel.

For the model structure part, adding more neurons is now to boost PINN's performance. It is good to compare all model structures based on the same number of parameters. Also, some better models' inferences are more costly. The authors should take the computational costs into account.

**Questions:**

Please explain your contribution: how is your model better than the survey on activations functions in PINNs: https://arxiv.org/abs/2209.02681?

How expensive/efficient is your model structure? Can your model outperform others under the same number of parameters.

---

> ### Author Response · Authors · 2023-11-20
> **Response to reviewer vRbU (part 1)**
>
> We thank the reviewer for their review and their questions. We apologize for taking time to post our rebuttal but the reviewer had asked questions that required us to undertake further experiments so as to answer the reviewers questions. We hope the reviewer will be understanding about this.
>
>
> **Activation functions have been well-studied in the PINN literature: https://arxiv.org/abs/2209.02681. Nowadays, proposing new activations does not seem to be novel.**
>
> We are not just proposing a new activation. It is clear the reviewer has not actually read the paper. As we mention in the introduction, Gaussian activations have already been proposed in the vision community. Yet there is not theoretical understanding why they perform so well. We give the first theoretical explanation through a mathematical formula as to why these activations are very good to use. We then test our theory in the area of PINNs.
>
> **Please explain your contribution: how is your model better than the survey on activations functions in PINNs: https://arxiv.org/abs/2209.02681?**
>
> Our contributions have clearly been explained on p.2 of the paper at the end of the introduction. The purpose of this paper is to lay a theoretical framework for using the NTK to choose an activation. We clearly show that by looking at the smallest eigenvalue scaling of a Gaussian activation and comparing this with results for other activations in the literature that training with a Gaussian activation should perform better than other networks. We then test this prediction on 4 partial differential equations and show superior results.
>
> In general, this is a theoretical paper one that is trying to bridge the gap between theory and practise in machine learning. The paper you have referenced: https://arxiv.org/abs/2209.02681, is an empirical paper that has simply run various experiments on different activations without actually providing any theoretical insight into why one activation performs better than others. We are the first to give a concrete mathematical explanation as to why using a Gaussian activation is better by proving how the minimum eigenvalue must scale. We are the first to have done this.
>
> **For the model structure part, adding more neurons is now to boost PINN's performance. It is good to compare all model structures based on the same number of parameters....**
>
> We compared all model structures with the same number of parameters. Did you read the paper? If you did you would see in experiment 5.1 and 5.2 all models are using the same number of parameters, 3 hidden layers, 128 neurons. In experiment 5.3 PINNsFormer and RFF use more parameters because this is what the original papers use.

---

> ### Author Response · Authors · 2023-11-20
> **Response to reviewer vRbU (part 2)**
>
> **How expensive/efficient is your model structure? Can your model outperform others under the same number of parameters.**
>
> The reviewer is clearly showing that they have not read the paper, which is extremely disappointing. Most of what the reviewer is asking is explained in the experiments section of the paper.
> All experiments were carried out with the same number of parameters. Except the RFF and PINNsFormer networks. For these two networks we used the standard networks used in the papers that introduced them. For experiment 5.1 we clearly write all networks were trained with 128 neurons and 3 hidden layers. This means all models had the same number of parameters. For experiment 5.2 we also clearly write that we trained all models with 3 hidden layers and 128 neurons. For experiment 5.3 we clearly write that we trained all networks with 3 hidden layers and 128 neurons except the PINNsFormer network which was trained with 9 hidden layers and 128 neurons as this is the architecture used in the original PINNsFormer papaer. The RFF network has a positional embedding layer of dimension 2 as this was what was used in the original RFF paper.
> Finally in experiment A.8 (in the appendix) we used two hidden layers with 128 neurons, except PINNsFormer and RFF used their standard architectures.
>
> Therefore, all networks have the same parameters in each experiments except the RFF and PINNsFormer networks which have more parameters. In all cases our networks do far better. Thus showing that our networks perform better than others with the same number of parameters and perform better than RFF and PINNsFormer with less parameters.
>
> To show the reviewer that our method is in fact superior at all levels of parameters we ran further experiments for different widths and depths. The following table show the train error in each setting. As can be seen the Gaussian PINN (ours) and EG-PINN (ours) performs the best at all parameter levels.
>
>
> Train results for Burgers' equation for fixed depth of 3 hidden layers and varying width. As can be seen from the results the Gaussian performs very well with all widths.
>
> |          | Tanh      | Gaussian | Sine     | Wavelet  |
> |----------|-----------|----------|----------|----------|
> | 32 width | $2.31e-2$ | $1.39e-2$| $1.31e-1$| $1.68e-1$|
> | 64 width | $1.09e-2$ | $5.43e-3$| $8.61e-2$| $9.14e-2$|
> |128 width | $9.86e-3$ | $1.61e-3$| $5.34e-2$| $5.92e-2$|
> |256 width | $8.11e-3$ | $1.32e-3$| $3.25e-2$| $4.21e-2$|
> |512 width | $7.01e-3$ | $1.11e-3$| $2.81e-2$| $3.88e-2$|
>
>
> Train results for Burgers' equation for fixed width of 128 neurons and varying depth (hidden layers). The Gaussian PINN does the best in this case too.
>
> |                 | Tanh      | Gaussian | Sine     | Wavelet  |
> |-----------------|-----------|----------|----------|----------|
> | 2 hidden layers | $1.13e-2$ | $5.51e-3$| $8.86e-2$| $9.68e-2$|
> | 3 hidden layers | $9.86e-3$ | $1.81e-3$| $5.34e-2$| $5.92e-2$|
> | 4 hidden layers | $8.13e-3$ | $1.59e-3$| $5.19e-2$| $5.37e-2$|
> | 5 hidden layers | $7.54e-3$ | $1.32e-3$| $5.17e-2$| $5.29e-2$|
> | 6 hidden layers | $7.45e-3$ | $1.21e-3$| $5.11e-2$| $5.12e-2$|
>
> Train results for high frequency diffusion equation for fixed depth of 3 hidden layers and varying width. You can see that for varying width EG-PINN performs the best.
>
> |              | G-PINN    | EG-PINN   | L-LAAF-PINN | RFF-PINN   | Tanh-PINN  |
> |--------------|-----------|-----------|-------------|------------|------------|
> | 32 width     | $1.02e-1$ | $5.56e-2$ | $9.96e-2$   | $8.98e3$   | $5.87e2$   |
> | 64 width     | $2.39e-2$ | $7.63e-3$ | $1.12e-2$   | $2.21e3$   | $9.76e1$   |
> | 128 width    | $8.61e-3$ | $2.23e-3$ | $5.89e-3$   | $8.44e2$   | $6.11e1$   |
> | 256 width    | $7.12e-3$ | $1.19e-3$ | $3.97e-3$   | $7.31e2$   | $5.89e1$   |
> | 512 width    | $7.06e-3$ | $9.98e-4$ | $3.12e-3$   | $6.67e2$   | $4.12e1$   |
>
> Train results for high frequency diffusion equation for fixed width of 128 neurons and varying depth (hidden layers). EG-PINN performs the best in this situation too.
>
> |                 | G-PINN    | EG-PINN   | L-LAAF-PINN | RFF-PINN   | Tanh-PINN  |
> |-----------------|-----------|-----------|-------------|------------|------------|
> | 2 hidden layers | $1.02e-2$ | $4.13e-3$ | $9.89e-3$   | $2.98e3$   | $1.87e2$   |
> | 3 hidden layers | $8.61e-3$ | $2.23e-3$ | $5.89e-3$   | $8.44e2$   | $6.11e1$   |
> | 4 hidden layers | $6.21e-3$ | $2.17e-3$ | $4.22e-3$   | $6.34e2$   | $5.83e1$   |
> | 5 hidden layers | $6.03e-3$ | $1.09e-3$ | $4.07e-3$   | $5.87e2$   | $5.13e1$   |
> | 6 hidden layers | $5.91e-3$ | $9.91e-4$ | $3.91e-3$   | $5.52e2$   | $4.98e1$   |
>
> If the reviewer feels these are important to include, we are happy to include them in the appendix. Though we feel most readers will simply skip reading such tables.

---

### Official Review · Reviewer_EzKm · 2023-10-31

**Soundness:** 3 good
**Presentation:** 3 good
**Contribution:** 3 good
**Rating:** 3
**Confidence:** 3

**Summary:**

This paper investigates a new architectural design for accelerating training processes of PINNs. There are largely two components proposed: (i) Gaussian activation functions and (ii) preconditioning of weight matrices in layers. The new design is largely inspired from the neural tangent kernel perspective of interpreting PINNs and the goal is to design architectures that increase the magnitude of eigenvalues of the NTK for faster convergence. The paper tests the proposed architecture on several benchmark partial differential equations.

**Strengths:**

- The main manuscript is well-written and easy to follow. Although the whole appendix has not been carefully examined, several parts of Appendix look correct, which gives the impression that the rest of the parts would be also credible.

- The idea of preconditioning the layer weights is interesting and two potential ideas were proposed (Jacobi and row equilibrated preconditioners), where both of them are diagonal matrices.

- The paper provides experimental results comparing the results with other baselines.

**Weaknesses:**

- One of the main findings in the previous work (Wang, et al, 2022, JCP) is that the discrepancy between the eigenvalues of NTKs computed from the data matching loss $L_b$ and the residual loss $L_r$ should be small in order to achieve faster convergence. Loss re-weighting shown in the paper (Eq. (8)) is to mitigate such imbalance between $L_b$ and $L_r$.

- In the paper, the proposed architectural change is only to improve minimization of boundary loss $L_b$ (as written in the fourth paragraph of page 4 and also in Theorem 3.1), which gives some concerns how this would affect the balance between $L_b$ and $L_u$. Would there be any theoretical explanation in this regard?

Concerns on practicality:
- As the method is derived from the NTK perspective, the users might have to assume that the width is larger than a certain value. On the other hand, as shown in the original PINNs, some empirical observations saying that making the neural network wider (e.g., larger than 50) would not provide much performance improvements.

- Regarding the wall time: based on the number of epochs, it does seem that the proposed algorithm is better. However, considering that the additional matrix-vector products are added to the computations, it might not be the case that the proposed method performs better in terms of efficiency measured in wall-time (or achievable accuracy given a wall time). Would there be any studies in this regard?

Experimental section is weak. Regarding this please see the questions below.

The presentation has some room to be improved

  - for example, Figure 3, the colorbar range can be fixed. Current, with the eyeball norm, every heatmap looks the same. Figure 4, the table inside does seem to require some height and width scaling.

  - Burger's => Burgers'

  - the authors may not follow the guidelines when it comes to referencing papers (\cite or \citep or else).

**Questions:**

Please refer to the question above.

Additional questions are:

-  could the authors comment on the efficiency or the scalability of the proposed method? For varying width and depth, the performance could be varying and some systematical assessment would be needed to really see the proposed method works better.

- how is the reported values are chosen? For example, in Table 3, could the authors can provide information on how were those numbers selected? Were the values generated by the single run of each model? were there hyper-parameter sweeps for baselines (width, depth, etc)? Is that particular specification used the experiments enough to assess the models' performance? (what are the justifications of that choice?)

- what is the definition of train error? is it the entire loss measured at training data instances? Also, in some tables, there are train error and in other tables, there are L2 train error. Are they different? How is the Rel. test error is defined?

---

> ### Author Response · Authors · 2023-11-20
> **Response to reviewer EzKm (part 1)**
>
> We thank the reviewer for their time for reviewing the paper and for the questions asked. We apologize for taking time to post our rebuttal but the reviewer had asked questions that required us to undertake further experiments so as to answer the reviewers questions. We hope the reviewer will be understanding about this.
>
> **In the paper, the proposed architectural change is only to improve minimization of boundary loss ......**
>
> This is a very good question and we thank the reviewer for asking it. We did allude to it in the "limitations" section where we concretely stated that one of the limitations of the work is that we were not able to obtain a scaling law for the pde residual term. What makes this difficult is that one needs to understand the theory of the NTK of derivatives of the network, which is much more harder than simply understanding the NTK of the network alone. Please have a look at the analysis in the appendix where it is shown even in the case of the NTK of the model how difficult the probability anylsis is. Extending this to the case of derivatives of the model is a very good question but is out of the scope of this work. So far, as we are aware no one in the machine learning literature has been able to obtain scaling laws for the NTK for derivatives of the network.
> However, there is still merit in our approach. As can be seen from experiment 1 applying a Gaussian not only influenced the boundary loss error but it also influenced the pde loss term. This suggest that the NTK of the pde term of the Gaussian network is also being affected in a positive way. This is only empirical though and as of yet we have no way to prove this theoretically. That being said research is incremental so it is important to understand that one cannot achieve everything in one go. We believe understanding how the NTK scaling is affected by the pde term is an important future work that the community should look into.
>
> **As the method is derived from the NTK perspective, the users might have to assume.....**
>
> This is always an issue with the NTK perspective. The NTK is actually only valid in the infinite width setting, for in this setting it was shown in the original NTK paper that the training under gradient descent converges to kernel regression given by a deterministic kernel which is the NTK. This means in practise it is impossible for there to be an equality between the NTK theory and practical implementations. The point of NTK theory is that it gives insight into how the dynamics of neural networks progress under gradient descent. The purpose of this paper is to mathematically prove a scaling law for the minimum eigenvalue of the NTK and use this scaling to obtain insight into the practical design of neural networks. The insight we are putting forward is that one should use the Gaussian activation. We then validate this insight empirically by showing in experiment 5.1 and 5.2 that the Gaussian performs far better than many other activations used within the literature. Furthermore in sec. A8 of the appendix of the original submission we have another experiment namely Poisson's equation, which has a periodic solution, and showed even in this case the use of a Gaussian activation leads to better results. In other words we tested the NTK part of our work on 3 different baseline experiments which is more than most papers on PINNs do. Making the network extremely wide has a disadvantage in that memory issues come up so generally in practise people only use the NTK theory for insight and try to find a comfortable width range that trains well. This is the approach that is taken in various papers in PINNs.
>
> Furthermore, the original NTK paper [1] has 2160 citations, yet it's results are only applicable in infinite width settings, which are impossible to test practically. The reason it has so many citations is because although it is a theoretical paper it still have provided the community with ample insight into how we should think about neural network training dynamics.
>
> [1] Neural Tangent Kernel: Convergence and Generalization in Neural Networks; 	arXiv:1806.07572 [cs.LG]

---

> ### Author Response · Authors · 2023-11-20
> **Response to reviewer EzKm (part 2)**
>
> **Regarding the wall time: based on the number of epochs, it does seem that the proposed algorithm is better. However, considering that the additional matrix-vector products are added to the computations....**
>
> We are somewhat surprised with the reviewers comment here as what they have said is not correct. We believe the reviewer is talking about the wall time of the Equilibrated PINN, where we use a Hessian vector product. We clearly show in experiment in 5.3 that the wall time (time is wall time in that experiment) is almost 4 times more than a standard Gaussian/Tanh PINN. Furthermore, we compared the wall time of all networks. We also have the Poisson equation experiment in section A8, that was included in the original submission, that also showed that the wall time of EG-PINN was much more when compared to standard PINNs. We further also included this as a limitation in the main paper. Thus this point has already been addressed with wall times given.
>
> **could the authors comment on the efficiency or the scalability of the proposed method? For varying width and depth, the performance could be varying and some systematical assessment would be needed.....**
>
> The proposed method works with almost all width and depth though as you go very deep and wide there is a memory issue. The other disadvantage of using large widths or large depths is that the time taken to train becomes much longer.
> We chose the width and depth in connection to the width and depth that is used in most PINN papers.
>
> We ran experiments on different width and depth configurations for you. Here are the results:
>
> Train results for Burgers' equation (experiment 5.1 in paper) for fixed depth of 3 hidden layers and varying width. As you can see Gaussian still performs the best:
>
> |          | Tanh      | Gaussian | Sine     | Wavelet  |
> |:--------:|:---------:|:--------:|:--------:|:--------:|
> | 32 width | $2.31e-2$ | $1.39e-2$| $1.31e-1$| $1.68e-1$|
> | 64 width | $1.09e-2$ | $5.43e-3$| $8.61e-2$| $9.14e-2$|
> |128 width | $9.86e-3$ | $1.61e-3$| $5.34e-2$| $5.92e-2$|
> |256 width | $8.11e-3$ | $1.32e-3$| $3.25e-2$| $4.21e-2$|
> |512 width | $7.01e-3$ | $1.11e-3$| $2.81e-2$| $3.88e-2$|
>
> Train results for Burgers' equation (experiment 5.1 in paper) for fixed width of 128 neurons and varying depth (hidden layers). Gaussian performs the best again:
>
> |                | Tanh      | Gaussian | Sine     | Wavelet  |
> |----------------|-----------|----------|----------|----------|
> | 2 hidden layers| $1.13e-2$ | $5.51e-3$| $8.86e-2$| $9.68e-2$|
> | 3 hidden layers| $9.86e-3$ | $1.81e-3$| $5.34e-2$| $5.92e-2$|
> | 4 hidden layers| $8.13e-3$ | $1.59e-3$| $5.19e-2$| $5.37e-2$|
> | 5 hidden layers| $7.54e-3$ | $1.32e-3$| $5.17e-2$| $5.29e-2$|
> | 6 hidden layers| $7.45e-3$ | $1.21e-3$| $5.11e-2$| $5.12e-2$|
>
> Train results for high frequency diffusion equation for fixed depth of 3 hidden layers and varying width. We see that EG-PINN does the best.
>
> |              | G-PINN    | EG-PINN   | L-LAAF-PINN | RFF-PINN   | Tanh-PINN  |
> |--------------|-----------|-----------|-------------|------------|------------|
> | 32 width     | $1.02e-1$ | $5.56e-2$ | $9.96e-2$   | $8.98e3$   | $5.87e2$   |
> | 64 width     | $2.39e-2$ | $7.63e-3$ | $1.12e-2$   | $2.21e3$   | $9.76e1$   |
> | 128 width    | $8.61e-3$ | $2.23e-3$ | $5.89e-3$   | $8.44e2$   | $6.11e1$   |
> | 256 width    | $7.12e-3$ | $1.19e-3$ | $3.97e-3$   | $7.31e2$   | $5.89e1$   |
> | 512 width    | $7.06e-3$ | $9.98e-4$ | $3.12e-3$   | $6.67e2$   | $4.12e1$   |
>
>
> Train results for high frequency diffusion equation for fixed width of 128 neurons and varying depth (hidden layers). EG-PINN outperforms all others.
>
>
> |                 | G-PINN    | EG-PINN   | L-LAAF-PINN | RFF-PINN   | Tanh-PINN  |
> |-----------------|-----------|-----------|-------------|------------|------------|
> | 2 hidden layers | $1.02e-2$ | $4.13e-3$ | $9.89e-3$   | $2.98e3$   | $1.87e2$   |
> | 3 hidden layers | $8.61e-3$ | $2.23e-3$ | $5.89e-3$   | $8.44e2$   | $6.11e1$   |
> | 4 hidden layers | $6.21e-3$ | $2.17e-3$ | $4.22e-3$   | $6.34e2$   | $5.83e1$   |
> | 5 hidden layers | $6.03e-3$ | $1.09e-3$ | $4.07e-3$   | $5.87e2$   | $5.13e1$   |
> | 6 hidden layers | $5.91e-3$ | $9.91e-4$ | $3.91e-3$   | $5.52e2$   | $4.98e1$   |
>
> If the reviewer feels these are important to include, we are happy to include them in the appendix. Though we feel most readers will simply skip reading such tables.

---

> ### Author Response · Authors · 2023-11-20
> **Response to reviewer EzKm (part 3)**
>
> **how is the reported values are chosen? For example, in Table 3, could the authors can provide information on how were those numbers selected?...**
>
> All networks were trained with the Adam optimizer using a learning rate of 1e-4. We found using a bigger learning rate made the Tanh network extremely unstable so opted for a learning rate that was stable for all networks so as to be fair. The training details are given in the appendix.
> In total we used 100 training points for the boundary/initial condition and 1000 training points for the pde residual. All points were sampled using a uniform distribution. The testing was then done by sampling 2000 points (uniformy) from the analytic solution.
> In table 3 the bndry./I.C. is train error and the pde error is also a training error. This means the
> the bndry./I.c. train error was computed over the 100 bndry./I.C. training points and the pde train error was computed over the 1000 sampled pde points.
>
> **what is the definition of train error? is it the entire loss measured at training data instances?....**
>
> The definition of training error is indeed the entire loss measure at training data. Note we are training full batch, as explained in the appendix, so it is the MSE error of the loss over the whole batch at each epoch. This is standard across the PINN literature.
>
> All errors are $L^2$ errors we apologize for calling some of them ``train" and other ``$L^2$ train". We have changed this.
>
> The $L^2$ rel. test error is the Mean square relative test error and is a standard definition in machine learning. Given the predicted solution $\widehat{u}$ and the true solution $u$. The relative test error over a collection of test points
> $\{x_i\}_{i=1}^N$ is defined:
>
> $$ \frac{1}{2N}\sum_{i=1}^N\frac{||\widehat{u}(x_i) - u(x_i)||^2_2}{||u(x_i)||^2_2}.$$
> In the case of a 1-dimensional output, which is our setting, the 2-norm is just the absolute value.
>
> We ran 5 trials and gave the average value the total number of trials. This was done to mitigate any issues that might arise during one run. The model width and depth we chose were taken from a width and depth sweep and consisted of the best results for all networks taking memory into account. As can be seen from the above results (see answer to your previous question) in the width and depth sweeps the G-PINN or the EG-PINN always performed best.
> Frequencies for the activations were also done by a sweep. Please see section A.4 of the appendix in the original submission where we clearly explained that we did a sweep to find the best hyperparameters for the activations.

---

> ### Author Response · Authors · 2023-11-20
> **Response to reviewer EzKm (part 4)**
>
> **for example, Figure 3, the colorbar range can be fixed. Current, with the eyeball norm, every heatmap looks the same....**
>
> This is why as a reviewer you should not eyeball a figure and actually pay attention to it. Furthermore, eyeballing does not constitute a norm. If you look at the table it clearly shows differences in the pressure. This is why looking at the numbers is important. If you zoom into the heat map you can see there are differences. For example look at the heap map of the sine network and the wavelet one. If you look at the middle right you can see two contour are discontinuous. If you compare this with the ground truth you can see they shouldn't be like that. Furthermore, if you zoom into the Gaussian network you can see it doesn't have this issue. This is all reflected in the test pressure values as well. Please don't eyeball a result and use that as justification as pointing out a weakness for a paper. It's bad reviewer practise to simply eye ball results of a paper you are reviewing.